# Confounder-Free Continual Learning via Recursive Feature Normalization

## Abstract

Confounders are extraneous variable that affect both the input and the target, resulting in spurious correlations and biased predictions. Learning feature representations that are invariant to confounders remains a significant challenge in continual learning. To remove the influence of confounding variables from intermediate feature representations, we introduce the Recursive Metadata Normalization (R-MDN) layer, which can be integrated into any stage within deep neural networks (DNNs). R-MDN performs statistical regression via the recursive least squares algorithm to maintain and continually update an internal model *state* with respect to changing distributions of data and confounding variables. Since R-MDN operates on the level of individual examples, it is compatible with state-of-the-art architectures like vision transformers. Our experiments demonstrate that R-MDN promotes equitable predictions across population groups, both within static learning and across different stages of continual learning, by reducing catastrophic forgetting caused by confounder effects changing over time.

## 1 Introduction

Confounders are extraneous variables that influence both the input and the target, resulting in spurious correlations that distort the true underlying relationships within the data (Greenland & Morgenstern, 2001; Ferrari et al., 2020). These spurious correlations introduce bias into learning algorithms, causing the feature representations learned by models, such as deep neural networks (DNNs), to be skewed (Buolamwini & Gebru, 2018; Obermeyer et al., 2019; Oakden-Rayner et al., 2020; Chen et al., 2021; Seyyed-Kalantari et al., 2020).

This problem is particularly prevalent in medical studies, such as those related to brain development (Casey et al., 2018), biological and behavioral health (Petersen et al., 2010; Brown et al., 2015), and dermatoscopic images (Tschandl et al., 2018), which are often confounded by demographic factors like age, sex, and socioeconomic background, and factors related to data acquisition. For example, a DNN trained to diagnose neurodegenerative disorders from brain MRIs might disproportionately rely on age instead of the underlying pathology. This may occur either due to the disease causing accelerated aging or in cases where there is a selection bias, i.e., having different distributions in the diseased cohort versus the control group. This can lead to models that are inequitable and inaccurate for certain populations (Rao et al., 2017; Seyyed-Kalantari et al., 2020; Zhao et al., 2020; Adeli et al., 2020b; Lu et al., 2021; Vento et al., 2022). Given these challenges, it is crucial to develop techniques that enable DNNs to focus on task-relevant features while remaining invariant to confounders, which are often available as auxiliary information or metadata in such datasets.

Methods such as BR-Net (Adeli et al., 2020a), MDN (Lu et al., 2021), P-MDN (Vento et al., 2022), and RegBN (Ghahremani Boozandani & Wachinger, 2024) have been previously proposed to address the challenges posed by confounders when training DNNs. There are, however, a multitude of situations where some of them cannot be applied, such as MDN, that requires estimating batch-level information, in association with vision transformers (Vaswani et al., 2017), and within the context of continual learning where one cannot look at future data for training. This *continuum* of data may arise in various contexts. For example, in a cross-sectional study (Tschandl et al., 2018), the training process is divided into distinct stages, with each stage featuring different data distributions. Conversely, in a longitudinal study (Petersen et al., 2010; Brown et al., 2015; Casey et al., 2018), the system does not have access to all data at the outset; instead, new data—such as patient visits

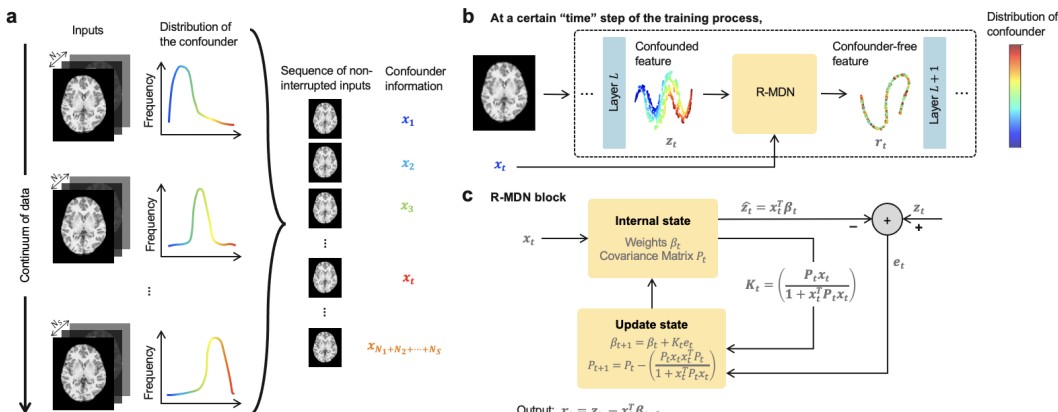

Figure 1: **A framework for confounder-free representations in continual learning. a.** A *continuum* of data with varying distributions of the confounder across different training stages can be viewed as a sequence of uninterrupted inputs that continually pass through a DNN. **b.** R-MDN is a layer that can be inserted at any stage within a DNN to remove the influence of the confounder from the intermediate feature representation. **c.** R-MDN performs a recursive step to update its internal *state* every time new data comes in.

in a clinical study—continually arrives over an extended period, often spanning several years. This creates a gap, as there is a need for algorithms that effectively and explicitly remove the influence of confounding variables under changing data or confounder distributions.

To this end, we propose *Recursive Metadata Normalization (R-MDN)* to remove (normalize) the effects of the confounding variables from the learned features of a DNN through statistical regression. Specifically, R-MDN leverages the recursive least squares (RLS) algorithm (Albert & Sittler, 1965), which has been widely utilized in adaptive filtering, control systems for reinforcement learning, and online learning scenarios (Xu et al., 2002; Gao et al., 2020). R-MDN is a layer that *can be inserted at any stage within a DNN*. The use of statistical linear regression is motivated by their success in de-confounding learned feature representations (McNamee, 2005; Brookhart et al., 2010; Pourhoseingholi et al., 2012; Adeli et al., 2018). The assumption of an underlying linear relationship between confounders and learned features arises from two key considerations: (1) decisions made by nonlinear models are often challenging to interpret, and (2) sufficiently powerful nonlinear models can extract almost any arbitrary variable from the information present in the features, even if those variables are not explicitly represented. R-MDN operates by iteratively updating its internal parameters—consisting of regression coefficients and an estimated inverse covariance matrix, which together form an internal model *state*—based on previously computed values whenever new data is received. This state represents the current understanding of the relationship between the learned features and the confounders, enabling the model to adapt dynamically as new data flows in. R-MDN, therefore, applies to static learning, where such a sequence of uninterrupted examples (minibatches) come from a single stationary distribution.

By design, a key advantage of R-MDN is in the context of continual learning, when each training stage consists of data drawn from different stationary distributions. This *continuum* of data can again be understood as a sequence of uninterrupted examples that a model learns from *over time*. Here, R-MDN does not need to train a stage-specific network. Instead, the internal state can be continuously updated over time as the model progresses through successive training stages. Therefore, only a single network equipped with R-MDN layers needs to be trained on the entire dataset, with the model being able to generalize across stages (data or confounder distributions)—both in performance and the ability to remove the effects of the confounders (see figure 1).

In summary, we propose R-MDN—a flexible normalization layer that is able to residualize the effects of confounding variables from learned feature representations of a DNN by leveraging the recursive least squares closed-form solution. It can do so under varying data or confounder distributions, making it an effective algorithm for both static and continual learning (sections 4.1, 4.2). We provide a theoretical foundation to our approach (section 3), and empirically validate it in different experimental setups and DNN architectures (sections 4.1.1, 4.1.2, 4.2.1, 4.2.2). We find that R-MDN helps in making equitable predictions for population groups (such as boys and girls) not only within a

single cross-sectional study (section 4.1.2), but also across different stages of training during continual learning, by minimizing catastrophic forgetting of confounder effects over time (section 4.2.2). Moreover, R-MDN generalizes well to examples where the influence from confounding variables is absent (section 4.2.1).

## 2 RELATED WORKS

Widely used techniques such as batch (Ioffe, 2015), layer (Ba et al., 2016), instance (Ulyanov, 2016), and group (Wu & He, 2018) normalization standardize intermediate feature representations of DNNs, i.e., they normalize them to have zero mean and unit standard deviation across different dimensions of the data. They do not explicitly remove the effects of confounding variables from these features.

Prior works have proposed methods for learning confounder-invariant feature representations based on domain-adversarial training (Liu et al., 2018; Wang et al., 2018; Sadeghi et al., 2019; Adeli et al., 2020a), closed-form statistical linear regression analysis (Lu et al., 2021), penalty-approach to gradient descent (Vento et al., 2022), regularization (Ghahremani Boozandani & Wachinger, 2024), disentanglement (Liu et al., 2021; Tartaglione et al., 2021), counterfactual generative modeling (Neto, 2020; Lahiri et al., 2022), fair inference (Baharlouei et al., 2020), and distribution matching (Baktashmotlagh et al., 2016; Cao et al., 2018). Among these, distribution matching techniques do not particularly remove the influence of individual confounders from learned features. Adversarial training, on the other hand, typically involves a confounder-prediction network applied to pre-logits feature representations, with an adversarial loss used to minimize the correlation between features and confounders. However, adversarial approaches struggle to scale effectively when faced with multiple confounding variables. Likewise, disentanglement, fair inference, and counterfactual generative modeling techniques only partially remove confounder effects from a single layer of the network (Zhao et al., 2020; Vento et al., 2022).

Among the methods listed earlier, Metadata Normalization (MDN) (Lu et al., 2021), which uses statistical regression analysis, is a popular technique. MDN is a layer that can be inserted into the DNN to residualize confounder effects from intermediate learned features. It does so through the ordinary least squares algorithm, wherein it computes a closed form solution for the expression $z = X\beta + r$ as $\beta = (X^\top X)^{-1} X^\top z$, where $z$ is the intermediate learned feature vector, $X$ is the confounder matrix, $\beta$ are the regression coefficients, and $r$ is the component in the learned features invariant to the confounder. To work with minibatches of data, MDN re-formulates the closed-form solution as $\beta = N\Sigma^{-1}\mathbb{E}(xz)$, where $\Sigma^{-1} = (X^\top X)^{-1}$ is pre-computed with respect to all training samples at the start of training, and the expectation $\mathbb{E}(xz)$ is computed using batch-level estimates during training. Not only does this pre-computation step require a space and computation overhead, employing batch-level statistics during training *precludes it from being used with vision transformers*, where computation is parallelized over individual examples. In the context of continual learning, where we might not have all data at the outset of training, MDN would have to repeatedly re-calculate $\Sigma^{-1}$ whenever new data comes in. Even if we did have all data at the outset, as in a cross-sectional study, a "look-ahead" operation would be required to have MDN compute $\Sigma^{-1}$ with respect to data from all stages of training.

To alleviate issues around the use of batch statistics, a penalty-approach to MDN (P-MDN) was proposed (Vento et al., 2022). The authors of P-MDN observe that MDN solves a bi-level nested optimization problem by having the network learn task-relevant features while also being invariant to the confounder. The authors suggest to solve a proxy objective $\min_{\beta,W} \mathcal{L}(\varphi(z - X\beta), y) + \gamma \mathcal{L}^*(z; X)$, where $\varphi$ is the non-linear computation to be performed within the network after the current layer, $y$ are the target labels, and $\gamma$ is a penalty parameter that trades off task learning with confounder-free feature learning. Now, P-MDN is able to work with arbitrary batch sizes. However, as we see in this work, $\gamma$ becomes very difficult to tune, and optimizing the proxy objective often leads to non-robust results with high variance across different seed runs.

For continual learning, methods such as those based on regularization (Kirkpatrick et al., 2017), knowledge distillation (Li & Hoiem, 2017), and architectural changes (Rusu et al., 2016; Mallya & Lazebnik, 2018; Bayasi et al., 2024) have been proposed to overcome *catastrophic forgetting*—the phenomenon where DNNs forget information learned in prior training stages when acquiring new

knowledge. Some of these methods are motivated by dealing with task (domain) specific biases by learning task (domain) general features (Arjovsky et al., 2019; Zhao et al., 2019; Creager et al., 2021). These methods, however, do not remove effects due to specific confounders from learned features. While domain-adversarial training and P-MDN still apply to the continual learning setting, we show in this paper that they do not perform well in many scenarios.

## 3 METHODOLOGY

Say we have $N$ training samples, where the input matrix $\boldsymbol{A} \in \mathbb{R}^{N \times d}$, for some dimension $d$, is associated with target labels $\boldsymbol{y} \in \mathbb{R}^N$ and information about the confounding variable $\tilde{\boldsymbol{x}} \in \mathbb{R}^N$. Let the output after a particular layer of a deep network be the features $\boldsymbol{z} \in \mathbb{R}^N$. Our goal is to obtain the residual $\boldsymbol{r}$ from the expression $\boldsymbol{z} = \tilde{\boldsymbol{x}}\tilde{\beta}_x + \boldsymbol{y}\tilde{\beta}_y + \boldsymbol{r} = \boldsymbol{X}\beta + \boldsymbol{r}$, where $\boldsymbol{X} = [\tilde{\boldsymbol{x}}\ \boldsymbol{y}]$ and $\beta = [\tilde{\beta}_x; \tilde{\beta}_y]$ is a set of learnable parameters. In other words, the learned features $\boldsymbol{z}$ are first projected onto the subspace spanned by the confounding variable and the labels, with the term $\tilde{\boldsymbol{x}}\tilde{\beta}_x$ corresponding to the component in $\boldsymbol{z}$ explained by the confounder and $\boldsymbol{y}\tilde{\beta}_y$ to that explained by the labels. We want to remove the influence of $\tilde{\boldsymbol{x}}$ from $\boldsymbol{z}$ while preserving the variance related to the labels. We thus compute the composite $\beta$ as explained below, but obtain the residual $\boldsymbol{r} = \boldsymbol{z} - \tilde{\boldsymbol{x}}\tilde{\beta}_x$; i.e., only with respect to $\tilde{\beta}_x$. This residual explains the components in the intermediate features irrelevant to the confounder but relevant to the labels, and thus for the classification task.

To accomplish this, we use the recursive least squares approach by modifying the closed-form solution obtained from having used an ordinary least squares (OLS) estimator instead:

$$\beta = \left(\sum_{i=1}^N X_{i,:} X_{i,:}^\top\right)^{-1} \left(\sum_{i=1}^N z_i X_{i,:}\right),\tag{1}$$

where $X_{i,:}$ is the $i^{\text{th}}$ row of $X$. If we represent $R(N) = \sum_{i=1}^N X_{i,:} X_{i,:}^\top$ and $Q(N) = \sum_{i=1}^N z_i X_{i,:}$, this is equivalent to writing $\beta = R(N)^{-1} Q(N)$.

Now, say that we have a new sample $A_{N+1,:}$ come in. The confounding variable and intermediate features for this sample are $X_{N+1,:}$ and $z_{N+1}$ respectively. This means that we need to compute new parameters

$$\beta' = R(N+1)^{-1} Q(N+1) = \left(R(N) + X_{N+1,:} X_{N+1,:}^\top\right)^{-1} \left(Q(N) + z_{N+1} X_{N+1,:}\right)\tag{2}$$

Fortunately, $R(N+1)^{-1}$ can be efficiently computed using the Sherman-Morrison rank-1 update rule (Sherman & Morrison, 1950):

$$\left(R(N) + X_{N+1,:} X_{N+1,:}^\top\right)^{-1} = R(N)^{-1} - \frac{R(N)^{-1} X_{N+1,:} X_{N+1,:}^\top R(N)^{-1}}{1 + X_{N+1,:}^\top R(N)^{-1} X_{N+1,:}},\tag{3}$$

We initialize $R(0)^{-1} = \epsilon \boldsymbol{I}$, where $\epsilon > 0$ is a small scalar, as most commonly used by prior works (Haykin, 2002; Stoica & Åhgren, 2002; Liu et al., 2009; Skretting & Engan, 2010). $\epsilon$ can be tuned to make the estimate approach that from using OLS (Stoica & Åhgren, 2002).

$\beta$ updates in RLS happen as shown in fig. 1c, with the derivation presented in suppl. A. An analysis of the computational and memory complexity is discussed in suppl. B.

### 3.1 MINI-BATCH LEARNING

We theorized R-MDN in an online learning setting above, with the system adapting to new information as it comes in one at a time. However, we can extrapolate this method to work with mini-batches of data, and be applicable to mini-batch learning when training the system. Let the system receive new mini-batches of information $\hat{\boldsymbol{X}} \in \mathbb{R}^{B \times d}$, for some batch size $B$, one after the other during training. Therefore, we can adapt the R-MDN approach to work with batches of new information by using the Sherman-Morrison-Woodbury formula (Woodbury, 1950):

$$\left(\boldsymbol{P} + \hat{\boldsymbol{X}}\hat{\boldsymbol{X}}^\top\right)^{-1} = \boldsymbol{P}^{-1} - \boldsymbol{P}^{-1}\hat{\boldsymbol{X}}\left(\boldsymbol{I} + \hat{\boldsymbol{X}}^\top \boldsymbol{P}^{-1}\hat{\boldsymbol{X}}\right)^{-1}\hat{\boldsymbol{X}}^\top \boldsymbol{P}^{-1}\tag{4}$$

### 3.2 FROM BATCH TO LAYER STATISTICS

Remember that one of the drawbacks of MDN is that it has to compute and store batch-level statistics $\Sigma$ with respect to the entire training data prior to training. Then, it uses this information along with computing batch-level estimates for each minibatch to residualize the features. The requirement of such batch-level estimates makes MDN unsuitable for modern SOTA architectures like vision transformers, wherein computations happen in parallel over all examples in a mini-batch. Incorporating an MDN module will inherently require an *aggregation* step for batch-level statistics to be computed, resulting in a significant computational overhead. R-MDN, on the other hand, operates on the level of individual examples in a minibatch. That is why it works in a purely online regime, as well as can be inserted in vision transformers to residualize intermediate learned features of the system.

### 3.3 REGULARIZATION

Notice that R-MDN can adapt quickly to changing data distributions over time due to its iterative nature, being especially helpful for continual learning where a *continuum* of data comes from several different stationary distributions. However, this iterative nature of the method might sometimes lead to it being too sensitive to small changes in the data. Random fluctuations, or data noise, can lead to unstable updates to R-MDN parameters. Therefore, we add a regularization term $\lambda \boldsymbol{I}$ to $P(N + B)$. $\lambda$ is a hyperparameter that is tuned during training (ablation in suppl. F). This has the effect of smoothing out the updates and stabilizing the residualization process, resulting in some robustness to noise. Additionally, adding this regularization term helps to ensure numerical stability by preventing the computation of an inverse for a matrix that might be singular or ill-conditioned.

## 4 EXPERIMENTAL RESULTS

### 4.1 STATIC LEARNING

First, we explore a static learning setting where the system only receives data from a single stationary distribution. In this setting, we test our methodology on two different binary classification tasks—a synthetic dataset that involves a continuous confounding variable (section 4.1.1), and a neuroimaging dataset for sex classification that contains a categorical confounder (section 4.1.2).

### 4.1.1 SYNTHETIC DATASET

We construct this dataset (after Adeli et al. (2020a); Lu et al. (2021)) by generating 2048 images of size $32 \times 32$, equally divided between two groups (categories). Each image consists of 3 Gaussian kernels: two on the main diagonal, i.e., quadrants II and IV, whose magnitudes are controlled by parameter $\sigma_A$, and one on the off-diagonal, i.e., quadrant III, whose magnitude is controlled by $\sigma_B$ (see figure 2). Differences in the distributions of $\sigma_A$ between the two groups are associated with the main effects (true discrimination cues) that should be learned by the system, whereas $\sigma_B$ is a confounding variable. An unbiased system will only use information from the main effects for categorization. Both $\sigma_A$ and $\sigma_B$ are sampled from the distribution $\mathcal{U}(1, 4)$ for group 1, and $\mathcal{U}(3, 6)$ for group 2. Since there is an overlap in the sampling range of

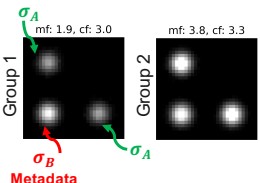

Figure 2: A sample from the synthetic dataset used for static learning.

the main effects between the two groups, the theoretical maximum accuracy that the system can achieve, were it to not depend on the confounding variable to make discrimination choices, would be $1 - \left(\frac{1}{2}\right) \mathcal{P}[\sigma_A \in \mathcal{U}(3, 4)] = 1 - \left(\frac{1}{2}\right) \left(\frac{1}{3}\right) = 0.833$.

We use a 2D convolutional neural network comprising of 2 stacks of convolutions and ReLU non-linearity, followed by 2 fully-connected layers. We apply residualization modules (either MDN, P-MDN, or R-MDN) after every convolution and pre-logits layers (other placement choices explored in suppl. G). During and after training, we quantify the high-dimensional non-linear correlation between the learned features from the pre-logits layer of the system and the confounding variable through the squared distance correlation ($dcor^2$) metric (Székely et al., 2007). A $dcor^2 = 0$ implies statistical independence between the two distributions.

Table 1: **Synthetic dataset results for static learning.** Absolute deviation from the theoretical accuracy $A$ ($\downarrow$) and squared distance correlation ($\downarrow$) for various methods and batch sizes. Results are shown over 100 runs of random model initialization seeds with a 95% confidence interval. Best results for each batch size are in bold. There is significant difference in all metrics across all batch sizes for different methods (one-way ANOVA $p < 10^{-58}$). Our method has a significantly better squared distance correlation than MDN for batch sizes less than 128 (post-hoc Tukey's HSD $p < 0.05$) and than P-MDN for batch sizes 2, 64, 256, and 1024 ($p < 0.05$).

| Method | Metric | Batch size | | | | |
|---|---|---|---|---|---|---|
| | | 2 | 16 | 64 | 256 | 1024 |
| Baseline | $\|bAcc - A\|$ | $10.49 \pm 0.037$ | $10.49 \pm 0.025$ | $10.50 \pm 0.023$ | $10.52 \pm 0.022$ | $10.55 \pm 0.029$ |
| | $dcor^2$ (group 1) | $0.408 \pm 0.002$ | $0.420 \pm 0.001$ | $0.421 \pm 0.001$ | $0.416 \pm 0.001$ | $0.310 \pm 0.005$ |
| | $dcor^2$ (group 2) | $0.388 \pm 0.003$ | $0.397 \pm 0.001$ | $0.394 \pm 0.001$ | $0.391 \pm 0.001$ | $0.281 \pm 0.005$ |
| MDN | $\|bAcc - A\|$ | $8.13 \pm 1.203$ | $4.93 \pm 0.424$ | $3.21 \pm 0.532$ | $\mathbf{0.52 \pm 0.335}$ | $0.95 \pm 0.335$ |
| | $dcor^2$ (group 1) | $0.977 \pm 0.010$ | $0.142 \pm 0.016$ | $0.086 \pm 0.010$ | $0.014 \pm 0.002$ | $\mathbf{0.003 \pm 0.000}$ |
| | $dcor^2$ (group 2) | $0.999 \pm 0.001$ | $0.046 \pm 0.009$ | $0.024 \pm 0.003$ | $\mathbf{0.000 \pm 0.001}$ | $\mathbf{0.000 \pm 0.000}$ |
| P-MDN | $\|bAcc - A\|$ | $4.65 \pm 0.448$ | $3.49 \pm 0.373$ | $1.85 \pm 1.151$ | $0.23 \pm 1.361$ | $1.58 \pm 1.983$ |
| | $dcor^2$ (group 1) | $0.042 \pm 0.013$ | $0.022 \pm 0.003$ | $0.050 \pm 0.007$ | $0.048 \pm 0.007$ | $0.098 \pm 0.020$ |
| | $dcor^2$ (group 2) | $0.060 \pm 0.021$ | $0.013 \pm 0.005$ | $0.015 \pm 0.002$ | $0.027 \pm 0.004$ | $0.091 \pm 0.026$ |
| R-MDN | $\|bAcc - A\|$ | $\mathbf{0.28 \pm 0.414}$ | $\mathbf{0.04 \pm 0.213}$ | $\mathbf{0.13 \pm 0.088}$ | $1.19 \pm 0.215$ | $\mathbf{0.19 \pm 0.296}$ |
| | $dcor^2$ (group 1) | $\mathbf{0.019 \pm 0.003}$ | $\mathbf{0.014 \pm 0.002}$ | $\mathbf{0.006 \pm 0.001}$ | $\mathbf{0.013 \pm 0.002}$ | $0.015 \pm 0.020$ |
| | $dcor^2$ (group 2) | $\mathbf{0.005 \pm 0.001}$ | $\mathbf{0.001 \pm 0.000}$ | $\mathbf{0.000 \pm 0.000}$ | $0.001 \pm 0.000$ | $0.008 \pm 0.017$ |

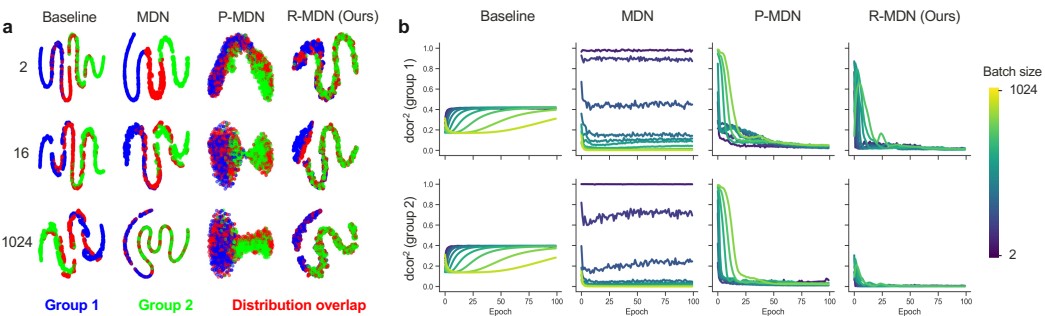

Figure 3: **Learned features and squared distance correlation. a.** t-SNE visualization of feature representations for different methods across batch sizes of 2, 16, and 1024. The more separable the distribution overlap $\mathcal{U}(3, 4)$ is in the feature space, the more the method relied on the confounder for discriminating between the groups. **b.** Squared distance correlation across batch sizes for different methods. Each curve represents a different batch size (ranging from 2 to 1024, in increments of powers of 2). Results are shown as the average over 100 runs of random model initialization seeds.

Our results are summarized in table 1. We observe that R-MDN consistently reaches the theoretically optimal accuracy and a lower $dcor^2$ across all batch sizes. The baseline "cheats" by making use of information from the confounding variable, resulting in a higher balanced accuracy and $dcor^2$. MDN is successful only for large batch sizes like 1024, while being significantly worse for small ones (Lu et al., 2021). While P-MDN is also able to remove the effects of the confounding variable from the learned features to a large extent across all batch sizes (as shown through a smaller $dcor^2$), the large variance across different seed runs suggests that the results are not consistent or robust.

A t-SNE visualization (Van der Maaten & Hinton, 2008) of the learned feature representations shows that the distribution overlap $\mathcal{U}(3, 4)$ for R-MDN is not separable from the two groups for all batch sizes, which means that the system does not use information from the confounding variable for categorization (figure 3a). In terms of convergence speed, R-MDN does significantly better than both MDN and P-MDN in removing the effects of the confounding variable from the learned features very quickly, especially for small batch sizes (figure 3b). This effect is attributable to fast convergence properties of the underlying RLS algorithm (Hayes, 1996; Haykin, 2002), and will be advantageous in a continual learning setting when we might not want to train a system until convergence on each training stage, but only for a single or few epochs (read suppl. J).

Table 2: **ABCD sex classification results for static learning.** Accuracy, true positive (TPR) and negative rates (TNR), difference between TPR and TNR, and squared distance correlation for both boys and girls for different methods. Results are shown as the mean and standard deviation over 5 folds of 5-fold cross validation, with data split by subject and site ID. Best and second-to-best results shown in bold and underlined respectively.

| Method | Accuracy | TPR | TNR | TPR - TNR | $dcor^2$ (boys) | $dcor^2$ (girls) |
|--------|----------|-----|-----|-----------|-----------------|------------------|
| Baseline | $86.86 \pm 0.354$ | $85.41 \pm 0.781$ | $88.32 \pm 0.770$ | $-0.029 \pm 0.016$ | $0.0127 \pm 0.0022$ | $0.0218 \pm 0.0029$ |
| Pixel-Space | $84.91 \pm 0.447$ | $83.04 \pm 2.900$ | $86.77 \pm 2.352$ | $-0.037 \pm 0.059$ | $0.0168 \pm 0.0041$ | $0.0239 \pm 0.0083$ |
| BR-Net | $81.63 \pm 0.499$ | $80.26 \pm 0.388$ | $83.01 \pm 0.908$ | $-0.027 \pm 0.011$ | $0.0127 \pm 0.0006$ | $0.0148 \pm 0.0002$ |
| MDN | $\mathbf{87.55 \pm 0.6630}$ | $\mathbf{87.43 \pm 3.301}$ | $87.66 \pm 4.277$ | $\underline{-0.002 \pm 0.084}$ | $0.0329 \pm 0.0140$ | $0.0624 \pm 0.0283$ |
| P-MDN | $86.41 \pm 0.876$ | $84.25 \pm 1.651$ | $\mathbf{88.57 \pm 1.540}$ | $-0.043 \pm 0.030$ | $\mathbf{0.0031 \pm 0.0009}$ | $\underline{0.0108 \pm 0.0017}$ |
| R-MDN | $85.08 \pm 0.591$ | $84.98 \pm 0.842$ | $85.18 \pm 1.125$ | $\mathbf{-0.002 \pm 0.018}$ | $\underline{0.0099 \pm 0.0029}$ | $\mathbf{0.0090 \pm 0.0027}$ |

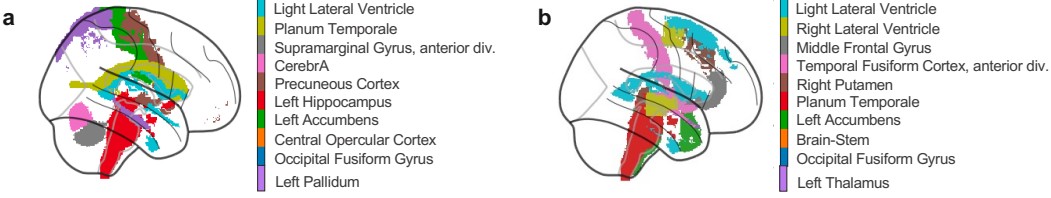

Figure 4: **Visualizing ROIs for ABCD sex classification.** The top 10 most relevant regions for distinguishing sex as determined by a model trained without and with R-MDN respectively.

### 4.1.2 ABCD SEX CLASSIFICATION

Next, we use T1-weighted structural MRIs from the ABCD (Adolescent Brain Cognitive Development) study (Casey et al., 2018) for the task of binary sex classification. Within a cross-sectional study setting, we take 10686 baseline (i.e., first visit) MRIs, confounded by scores from the Pubertal Development Scale (PDS)—a validated measure of pubertal stage identified through self-assessment. PDS is a confounder because it is larger in girls ($2.175 \pm 0.9$) than in boys ($1.367 \pm 0.6$) in this study, and statistically significant (read suppl. C; Adeli et al. (2020b)). PDS categorizes participants as either (1) pre-pubertal, (2) early-pubertal, (3) mid-pubertal, (4) late-pubertal, or (5) post-pubertal (Carskadon & Acebo, 1993).

We start with a 3D CNN as the base model, consisting of three stacks of convolutional layers, each followed by ReLU non-linearity and max pooling, and ending with two fully connected layers. As before, we insert a residualization module after every layer except the last one. In addition to this approach, we establish two additional baselines—one where we use BR-Net, and adversarial training framework, with the same base model as the encoder, and another where we pre-process the input images prior to training by regressing out the influence of confounders directly from the pixel space (hereafter referred to as Pixel-Space Residualization). We set the batch size to be 128, which is the largest that can be fit in GPU memory (see additional details in suppl. D).

We observe that the base model has a high accuracy, but at the cost of being significantly biased towards girls—its feature representations have a larger $dcor^2$ with the confounder for girls than boys, and it also has a higher true negative rate (table 2). This is because the base model makes use of the pubertal development scores to drive its predictions. On the other hand, R-MDN incurs a modest decrease in performance, but significantly drives down the correlation between the learned features and the confounder for both boys and girls. Moreover, it has the lowest mean difference between the true positive and true negative rates among all evaluated methods (quantified in table 2, visualized in figure 9a,b,c), signifying that it is not biased toward children of either sex. Other methods like MDN and P-MDN have a higher prediction accuracy, but either fail to drive down the correlation between the features and the confounder due to requiring relatively larger batch sizes, or remain biased towards girls despite driving the correlation down.

Further validation of R-MDN in learning confounder-free feature representations is revealed when the model does not use the cerebellum—which is the region mostly confounded by PDS (Adeli et al., 2020b)—for distinguishing sex, when the base model does (figure 4a,b).

Table 3: **Quantifying metrics for the synthetic dataset in continual learning.** BWTd and FWTd mean and standard deviation for different methods and datasets. The closer the bar to 0, the better the model. A total of 5 runs were performed with different model initialization seeds.

| Method | Dataset 1 Confounder dist. changes | | Dataset 2 Main effects change | | Dataset 3 Both distributions change | |
|---|---|---|---|---|---|---|
| | BWTd ($\times 10^{-2}$) | FWTd | BWTd ($\times 10^{-2}$) | FWTd | BWTd ($\times 10^{-2}$) | FWTd |
| Baseline | **0.032 ± 0.041** | 0.191 ± 0.000 | **-0.051 ± 0.130** | 0.210 ± 0.000 | -0.371 ± 0.018 | 0.314 ± 0.000 |
| BR-Net | -1.278 ± 1.373 | 0.052 ± 0.038 | -1.428 ± 1.810 | 0.044 ± 0.026 | -0.552 ± 0.562 | 0.082 ± 0.043 |
| P-MDN | -0.608 ± 1.367 | 0.040 ± 0.014 | -0.759 ± 2.460 | 0.047 ± 0.007 | -1.372 ± 2.209 | 0.056 ± 0.010 |
| R-MDN | 0.151 ± 0.140 | **0.024 ± 0.009** | 0.066 ± 2.170 | **0.026 ± 0.004** | **-0.039 ± 0.775** | **0.024 ± 0.003** |

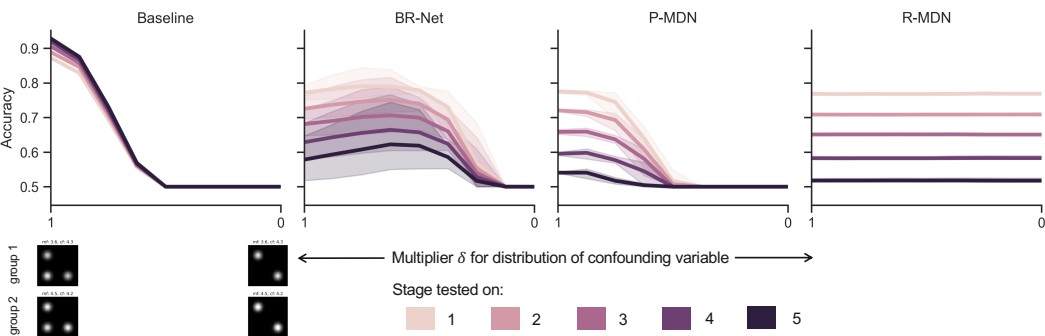

Figure 5: **Effects of the presence of the confounder for task generalization.** Accuracy as a function of the change in the intensity of the confounder, from 1 (completely present) to 0 (completely absent). All results shown as the mean and 95% CI over 5 runs.

## 4.2 CONTINUAL LEARNING

Here, we move from training on data sampled from a single stationary distribution to training on a *continuum* of data by slightly modifying the setting described by Lopez-Paz & Ranzato (2017): Given a 4-tuple $(a_i, x_i, y_i, s_i)$ for $i \in [N]$, where $a_i \in \mathcal{A}$ is the input, $x_i \in \mathcal{X}$ is the confounder, $y_i \in \mathcal{Y}$ is the label, and $s_i \in \mathcal{S}$ is the training stage descriptor, it satisfies *local iid*, i.e., $(a_i, x_i, y_i) \overset{\text{iid}}{\sim} \mathcal{P}_{s_i}(\mathcal{A}, \mathcal{X}, \mathcal{Y})$. Our goal is to learn a classifier $g : \mathcal{A} \times \mathcal{S} \to \mathcal{Y}$, that is able to predict the label $y$ associated with an unseen 2-tuple $(a, s)$, where $(a, y) \sim \mathcal{P}_s$ *at any point* during or after training on the $\mathcal{S}$ stages, in a way such that it does not make use of confounder information in $x$, if any.

### 4.2.1 A CONTINUUM OF SYNTHETIC DATASETS

We transform the synthetic data from section 4.1.1 into a *continuum* of data with varying distributions of the confounding variable and main effects. Specifically, we design 3 different datasets, each with 5 stages of training that arrive sequentially. For stage 1 across all 3 datasets, we start out with the parameter controlling the main effects $\sigma_A \in \mathcal{U}(3, 5)$ for group 1, and $\in \mathcal{U}(4, 6)$ for group 2. We use the same distributions for $\sigma_B$ (that controls the magnitude of the confounding variable). This means that predictions kernels (i.e., those associated with true discrimination cues) are more "intense" (have a higher magnitude) for images in group 2 than in group 1. Over "time", the distributions of either the confounding variable, main effects, or both change in a way that emphasize biased learning of a classifier that makes uses of confounder information for discrimination. A complete description of the 3 datasets is presented in suppl. C.

To quantify knowledge transfer, we define the Backward Transfer distance (BWTd) and Forward Transfer distance (FWTd) metrics. These metrics are adapted from BWT and FWT defined in Lopez-Paz & Ranzato (2017) to work with the setting where a model is expected to achieve certain theoretical accuracies on data from both previous and future stages of training. Say we have a total of $S$ stages. Let $R_{i,j}$ denote the classification accuracy of the model on stage $s_j$ after learning stage $s_i$. And let $\boldsymbol{A}_i$ denote the theoretical maximum accuracy for stage $s_i$. Then,

$$\text{BWTd} = \frac{1}{S-1} \sum_{i=1}^{S-1} |R_{S,i} - \boldsymbol{A}_i| - |R_{i,i} - \boldsymbol{A}_i| \tag{5}$$

$$\text{FWTd} = \frac{1}{S-1} \sum_{i=2}^{S} |R_{i-1,i} - \boldsymbol{A}_i| \tag{6}$$

The smaller these metrics, the better the model. While all methods are very good at backwards transfer, R-MDN is better at forward transfer as well (table 3 and figure 8b). This means that even with changing distributions of the confounding variable, R-MDN only "looks at" the main effects for classification, allowing it to learn features that transfer well to later tasks while remaining invariant to the confounder itself. Other methods make use of confounder information to various degrees, pulling their classification accuracy away from $\boldsymbol{A}$. This is also reflected in R-MDN consistently achieving an accuracy near the theoretical maximum for the test sets of each stage, while also showing the lowest correlation with the confounding variable (figure 8c).

We also quantify how different methods generalize to unseen images where the confounder is absent (figure 5). This issue arises in situations where, for example, a model is trained on data from multiple hospitals, with the machine type acting as a confounder, and then tested on data from a completely different hospital with a uniform machine type (Zech et al., 2018). In this scenario, the base model experiences a sharp drop in performance when the distribution of the confounder changes in the test data. Both BR-Net and P-MDN show some resistance to the distribution shift but fail when the confounder is entirely absent. In contrast, R-MDN maintains consistent performance across all distributions.

### 4.2.2 HAM10000 Dermatoscopic Skin Lesion Classification

Lastly, we use the HAM10000 dataset to classify 2D dermatoscopic images of pigmented skin lesions into seven distinct diagnostic categories (Tschandl et al., 2018). The dataset consists of 10015 images, which we divide into five training stages. In each stage, the age distribution—the confounding variable for this study (read suppl. C)—varies, with younger populations represented in the earlier stages and older populations in the later stages. For each stage, we randomly allocate 80% of the images for training and the remaining 20% for evaluation.

In this experiment, we transition from a CNN to a vision transformer as the base architecture, and as the encoder for BR-Net. For R-MDN, we explore three different variants: (A) inserting the R-MDN layer after the self-attention layer in every transformer block, as well as after the pre-logits layer; (B) inserting it at the end of every transformer block and after the pre-logits layer; and (C) inserting it only after the pre-logits layer. For P-MDN, we place the P-MDN layer right after the pre-logits layer. Additionally, we establish three continual learning frameworks as baselines: elastic weight consolidation (EWC) (Kirkpatrick et al., 2017) as a regularization method, learning without forgetting (LwF) (Li & Hoiem, 2017) for knowledge distillation, and PackNet (Mallya & Lazebnik, 2018), an architectural method that applies iterative pruning.

Results are summarized in table 4 and visualized in figure 10. While the base model performs decently on the classification task, it exhibits significantly lower backward transfer on earlier stages of training. In contrast, R-MDN not only effectively removes the confounder's influence from the features, as indicated by a low $dcor^2$ value, but also demonstrates significantly better backward transfer than the base model. We try to understand why this is by looking at the t-SNE visualizations of their features (figure 6). When the base model is trained on the final stage, it learns to clearly separate feature clusters for each of the seven diagnostic categories. However, it is possible that this separation is influenced by the stage-specific distribution of the confounding variable, leading to spurious correlations driving cluster separation, and thus poor transfer to previous data. On the other hand, R-MDN also forms feature clusters for the different categories but without introducing the same level of separation. This might be because R-MDN relies on task-relevant information, rather than the confounder, to discriminate between the categories. As a result, R-MDN is able to apply the knowledge learned in the current stage to previous stages of training, improving its overall backward transfer performance. Such backward transfer seems logical, since the classification task is the same across all stages, and only the confounder distribution changes.

Table 4: **HAM10K skin lesion classification results for continual learning.** Results shown as the mean and standard deviation over test sets of different stages of training for the model after being trained on the last training stage. Best and second-to-best results shown in bold and underlined respectively.

| Method | Accuracy | Average dcor$^2$ | BWT | FWT |
|---|---|---|---|---|
| Baseline | $\underline{0.7095 \pm 0.0626}$ | $0.0864 \pm 0.0336$ | $0.0278 \pm 0.0446$ | $\underline{0.5125 \pm 0.0705}$ |
| BR-Net | $\mathbf{0.7247 \pm 0.0627}$ | $\underline{0.0544 \pm 0.0534}$ | $-0.0207 \pm 0.0166$ | $\mathbf{0.5592 \pm 0.0897}$ |
| P-MDN | $0.6750 \pm 0.0945$ | $0.2595 \pm 0.0620$ | $\underline{0.0706 \pm 0.0622}$ | $0.4391 \pm 0.0372$ |
| R-MDN (A) | $0.5503 \pm 0.0541$ | $0.0928 \pm 0.0630$ | $-0.0268 \pm 0.0248$ | $0.4130 \pm 0.0709$ |
| R-MDN (B) | $0.5288 \pm 0.0571$ | $0.0739 \pm 0.0555$ | $0.0571 \pm 0.0693$ | $0.3362 \pm 0.0881$ |
| R-MDN (C) | $0.6919 \pm 0.0723$ | $\mathbf{0.0475 \pm 0.0247}$ | $\mathbf{0.1246 \pm 0.2123}$ | $0.3997 \pm 0.1555$ |
| EWC | $0.6437 \pm 0.0586$ | $0.0938 \pm 0.0506$ | $0.0698 \pm 0.0238$ | $0.4457 \pm 0.0620$ |
| EWC + R-MDN (C) | $0.6739 \pm 0.0686$ | $0.0592 \pm 0.0488$ | $0.0754 \pm 0.1614$ | $0.4404 \pm 0.1305$ |
| LwF | $0.7356 \pm 0.0757$ | $0.0512 \pm 0.0407$ | $0.0387 \pm 0.0390$ | $0.5277 \pm 0.0605$ |
| LwF + R-MDN (C) | $0.7186 \pm 0.0736$ | $0.0354 \pm 0.0210$ | $0.1348 \pm 0.1994$ | $0.4434 \pm 0.1403$ |
| PackNet | $0.6849 \pm 0.0745$ | $0.0470 \pm 0.0304$ | $0.0538 \pm 0.0670$ | $0.4965 \pm 0.0611$ |

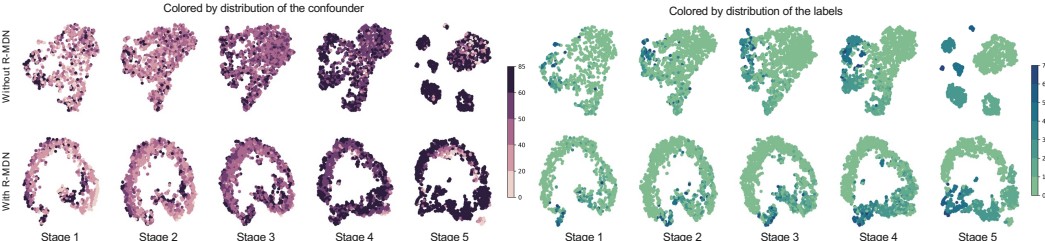

Figure 6: **Visualizing learned features for HAM10K skin lesion classification.** t-SNE representation of the learned features after training a model with and without R-MDN on all stages of continual learning.

Out of the three R-MDN variants, R-MDN (C) performs the best. Moreover, applying R-MDN to classic continual learning frameworks like EWC and LwF still drive the correlation with the confounder significantly down. In contrast, other methods like BR-Net and P-MDN do not perform as well. BR-Net catastrophically forgets past information, and P-MDN fails to effectively remove confounder effects.

## 5 CONCLUSION

In this work, we presented Recursive Metadata Normalization (R-MDN)—a flexible layer that can be inserted at any stage within deep neural networks to remove the influence of confounding variables from feature representations. R-MDN leverages the recursive least squares algorithm to operate at the level of individual examples, enabling it to adapt to changing data and confounder distributions in continual learning. It also promotes equitable outcomes across population groups and mitigates the catastrophic forgetting of confounder effects over time. As a direction for future work, R-MDN could be adapted and evaluated on datasets beyond medical contexts, such as video streams and audio signals, where confounding variables like environmental noise, lighting conditions, camera angles, or speaker accents might introduce spurious correlations in the data and bias the learning algorithm.

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

# Appendix

## Table of Contents

## A    DERIVING PARAMETER UPDATES FOR R-MDN

We know that the closed-form solution of OLS is

$$\beta = \left( \sum_{i=1}^{N} X_{i,:} X_{i,:}^{\top} \right)^{-1} \left( \sum_{i=1}^{N} z_i X_{i,:} \right) = R(N)^{-1} Q(N) \tag{7}$$

Firstly,

$$Q(N+1) = Q(N) + z_{N+1} X_{N+1,:} \tag{8}$$

Additionally, using the Sherman-Morrison rank-1 update rule,

$$R(N+1) = \left( R(N) + X_{N+1,:} X_{N+1,:}^{\top} \right)^{-1} = R(N)^{-1} - \frac{R(N)^{-1} X_{N+1,:} X_{N+1,:}^{\top} R(N)^{-1}}{1 + X_{N+1,:}^{\top} R(N)^{-1} X_{N+1,:}} \tag{9}$$

Let

$$P(N+1) = R(N+1)^{-1} = P(N) - K(N+1) X_{N+1,:}^{\top} P(N), \tag{10}$$

where the Kalman Gain

$$K(N+1) = \frac{P(N) X_{N+1,:}}{1 + X_{N+1,:}^{\top} R(N)^{-1} X_{N+1,:}} \tag{11}$$

Rewriting eq. 11,

$$K(N+1) \left[ 1 + X_{N+1,:}^{\top} P(N) X_{N+1,:} \right] = P(N) X_{N+1,:}$$
$$K(N+1) + K(N+1) X_{N+1,:}^{\top} P(N) X_{N+1,:} = P(N) X_{N+1,:}$$
$$K(N+1) = \left[ P(N) - K(N+1) X_{N+1,:}^{\top} P(N) \right] X_{N+1,:}$$
$$K(N+1) = P(N+1) X_{N+1,:} \quad \text{[using eq. 10]} \tag{12}$$

Finally,

$$\begin{aligned}
\beta(N+1) &= P(N+1)Q(N+1) \\
&= P(N+1)Q(N) + P(N+1)z_{N+1}X_{N+1,:} \quad \text{[using eq. 8]} \\
&= \left[ P(N) - K(N+1)X_{N+1,:}^{\top}P(N) \right] Q(N) + P(N+1)z_{N+1}X_{N+1,:} \quad \text{[using eq. 10]} \\
&= \left[ P(N) - K(N+1)X_{N+1,:}^{\top}P(N) \right] Q(N) + K(N+1)z_{N+1} \quad \text{[using eq. 11]} \\
&= P(N)Q(N) + K(N+1) \left[ z_{N+1} - X_{N+1,:}^{\top}P(N)Q(N) \right] \quad \text{[using eq. 11]} \\
&= \beta(N) + K(N+1) \left[ z_{N+1} - X_{N+1,:}^{\top}\beta(N) \right] \quad \text{[using eq. 7]} \\
&= \beta(N) + K(N+1)e(N+1),
\end{aligned} \tag{13}$$

where $e(N+1) = z_{N+1} - X_{N+1,:}^{\top}\beta(N)$, the a priori error computed before we update residual model parameters $\beta$.

## B   COMPUTATIONAL AND MEMORY COMPLEXITY

MDN, P-MDN, and R-MDN each have their tradeoffs in terms of computational complexity, memory complexity, and the extent to which the influence of the confounder is removed from the learned feature representations. As demonstrated in this work, R-MDN empirically works better than both MDN and P-MDN. The asymptotic complexity of each is presented here.

Say there are $N$ training examples, broken into batches of size $B$. Let the confounder matrix $X$ have a shape $N \times p$, where $p$ is associated with the number of confounders, the target, and a bias of 1, and the intermediate learned feature representations have a size of $N \times h$.

Firstly, MDN internally uses the linear least squares estimator, which requires pre-computing the matrix $\Sigma = X^\top X$ in $\mathcal{O}(Np^2)$ steps. Inverting this $p \times p$ matrix further requires $\mathcal{O}(p^3)$ steps. Then, for every batch of information during training, a batch level estimate $\overline{X}^\top \overline{z}$ is produced, where the $\overline{(\cdot)}$ operation refers to a batch instead of the entire training data. This takes $\mathcal{O}(Bph)$ steps. Post-multiplying this $p \times h$ matrix with $\Sigma^{-1}$ requires $\mathcal{O}(p^2 h)$ steps. If computations over batches of information occur $E$ times, the total computational complexity becomes $\mathcal{O}(p^3 + Np^2 + E(p^2 h + Bph))$. In terms of memory complexity, a $p \times p$ $\Sigma^{-1}$ needs to be stored, along with the residual model parameters $\beta$ of size $p \times h$.

For R-MDN, computations only occur over batches of information. In memory, residual model parameters $\beta$ of size $p \times h$ and an estimate of the inverse covariance matrix $P$ of size $p \times p$ are required. For every processing iteration, computing the Kalman gain $K$ requires $\mathcal{O}(Bp^2)$ steps for $PX^\top$, $\mathcal{O}(B^2 p + Bp^2)$ steps for $XPX^\top$, $\mathcal{O}(B^3)$ for inverting this latter matrix, and $\mathcal{O}(B^2 p)$ steps for multiplying the matrices together. Updating $P$ using $KX$ requires $\mathcal{O}(B^2 p)$ steps. And finally, updating $\beta$ requires computing $Ke$ in $\mathcal{O}(Bph)$ steps. The total computational complexity turns out to be $\mathcal{O}(E(B^3 + B^2 p + Bp^2 + Bph))$ steps. Empirically, R-MDN works best with small batch sizes $B$, showing very fast convergence rates, and having a computational complexity that is independent of the size of the training dataset. This becomes important for continual learning, especially longitudinal studies, where data collected over several years or decades can prohibit the use of MDN.

P-MDN does not use a closed-form solution to linear statistical regression. Instead it uses gradient descent to optimize a proxy objective. Thus, the only memory complexity stems from storing $\beta$ parameters of size $p \times h$. The computational complexity is dominated by the number of iterations required to navigate the proxy loss landscape, with results that are not often robust with high variance across runs.

# C    ADDITIONAL DETAILS ON DATASETS

## C.1    ABCD STUDY

The Adolescent Brain Cognitive Development (ABCD) study (https://abcdstudy.org) is a multisite, longitudinal study. More than 10,000 boys and girls from the U.S. between the ages of 9-10 were recruited based on a diversity of races and ethnicities, education and income levels, and living environments (Thompson et al., 2019). See Garavan et al. (2018) for a more detailed account of the population neuroscience approach to recruitment and inclusion/exclusion criteria. Appropriate consent was requested before participation in the ABCD study. Data is anonymized and curated, and is released annually to the research community through the NIMH Data Archive (see data sharing information at https://abcdstudy.org/scientists/data-sharing/). The ABCD data repository grows and changes over time. The ABCD data used in this report came from DOI 10.15154/8873-zj65.

Table 5 shows the distribution of participants (boys and girls) in the study with respect to age, pubertal development score (PDS), and race. PDS is significantly larger for girls than boys, and thus serves as a confounder for this study.

Table 5: **Variable distributions across boys and girls in the ABCD study**. Mean and standard deviation for age and pubertal development scale (PDS), and the number of subjects of each race in the study across boys and girls. PDS is an integer between 1-5. Differences are significant across boys and girls for age and PDS (measured using a two-sample t-test) but not race. Girls have a higher PDS than boys in the study. All values are for the first visit of each subject.

|  |  | **Boys** | **Girls** | **p-value** |
|---|---|---|---|---|
| Age (in months) |  | $119.17 \pm 7.563$ | $118.81 \pm 7.520$ | $<0.001$ |
| PDS |  | $1.367 \pm 0.615$ | $2.175 \pm 0.904$ | $<0.001$ |
| Race | White | 5954 | 5370 |  |
|  | Black | 1510 | 1558 |  |
|  | Hispanic | 2186 | 2084 | NS |
|  | Asian | 214 | 232 |  |
|  | Other | 1162 | 1090 |  |

## C.2    A CONTINUUM OF SYNTHETIC DATASETS

**Dataset 1: Confounding variable distribution changes.** We keep the distribution of main effects constant but vary that of the confounding variable across different stages. With every new stage, we decrease the entire range of $\sigma_B$ by 0.125 for group 1, and increase it by the same amount for group 2. For an unbiased classifier that uses short-cut learning by focusing on the confounder distribution, the problem becomes easier with "time" and performance will likely increase. This is what we observe with the baseline model, which has the same architecture as that in section 4.1.1 (see figure 8c).

**Dataset 2: Distribution of main effects changes.** Next, we keep the distribution of the confounding variable constant and vary that of the main effects across different stages instead. In contrast to the above, with every new stage, we increase the entire range of $\sigma_A$ by 0.125 for group 1, and decrease it by the same amount for group 2. This results in the problem becoming more difficult with "time". Performance for an unbiased classifier should drop for later tasks during learning.

**Dataset 3: Distributions of both the confounding variable and main effects change.** This dataset is a combination of the above two, with the difference in the distribution of the confounding variable across the two groups becoming more pronounced with "time", while that of the main effects starting to become more similar.

## C.3    HAM10000 DATASET

The *Human Against Machine with 10000 training images* (HAM10000) dataset (Tschandl et al., 2018) is a multi-source collection of 10015 dermatoscopic images for diagnosis of common pig-

mented skin lesions. These images have been collected from different populations through different modalities. Diagnostic categories include:

- **akiec:** actinic keratoses and intraepithelial carcinoma or Bowen's disease
- **bcc:** basal cell carcinoma
- **bkl:** benign keratosis-like lesions (solar lentigines or seborrheic keratoses and lichen-planus like keratoses)
- **df:** dermatofibroma
- **mel:** melanoma
- **nv:** melanocytic nevi
- **vasc:** vascular lesions (angiomas, angiokeratomas, pyogenic granulomas and hemorrhage)

Lesions were confirmed either through histopathology, follow-up examinations, expert consensus, or in-vivo confocal microscopy.

Figure 7 shows the distribution of age for various diagnostic categories. The change in the age distribution is significant. Age is a confounder for this dataset because it affects both the target categories (certain categories like melanoma mostly occur in older patients) and the input images (skin appearance might change with age).

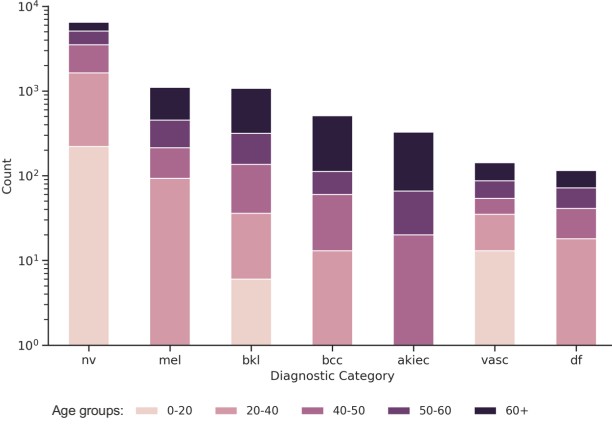

Figure 7: Distribution of dermatoscopic images across different diagnostic categories for various age brackets in the HAM10000 dataset.

# D  ADDITIONAL DETAILS ON METHODS

All experiments were run on a single NVIDIA GeForce RTX 2080 Ti with 11GB memory size and 8 workers on an internal cluster.

## D.1  SYNTHETIC DATASET

The base model was a CNN consisting of two convolutional layers followed by two fully connected layers. The first convolutional layer had 16 output channels and a kernel size of 5, the second had 32 output channels and a kernel size of 5, and the pre-logits fully connected layer had a hidden size of 84. Models were trained for 100 epochs with different batch sizes. Parameters of the R-MDN model were optimized using Adam (Kingma & Ba, 2014), with a learning rate initialization of 0.0001 that decayed by 0.8 times every 20 epochs. The regularization parameter for R-MDN was set to 0.0001.

## D.2  ABCD STUDY

Raw MRI images were downloaded, skull-stripped, and affinely registered to the MNI 152 template (Mazziotta et al., 1995; 2001a;b). Data augmentation involved removing MRIs for all subjects that did not have an associated PDS score recorded. We downscaled all MRIs to $64 \times 64 \times 64$ volumes, performed random one voxel shift and one degree rotation in all three Cartesian directions, and random left-right flip (since sex affects the brain bilaterally (Hill et al., 2014; Hirnstein et al., 2019)) for training images. To evaluate the models, we perform 5 runs of 5-fold cross validation across different model initialization seeds, with images split by subject and site ID, and having approximately an equal number of boys and girls in each fold.

The base model was a CNN consisting of three convolutional layers, each followed by max pooling, and two fully connected layers. The first convolutional layer had 8 output channels with a kernel size of 3, the second had 16 output channels with a kernel size of 3, and the third had 32 output channels with a kernel size of 3. The pre-logits fully connected layer had a hidden size of 32. For max pooling, the first and second layers used a kernel size of 2 with a stride of 2, while the third layer had a kernel size of 4 with a stride of 4. Models were trained for 50 epochs with a batch size of 128. Parameters of the R-MDN model were optimized using Adam, with a learning rate initialization of 0.0005 that decayed by 0.7 times every 4 epochs. The regularization parameter for R-MDN was set to 0.

## D.3  A CONTINUUM OF SYNTHETIC DATASETS

Models were trained for 100 epochs with a batch size of 128. Parameters of the R-MDN model were optimized using Adam, with a learning rate initialization of 0.0005 that decayed by 0.8 times every 20 epochs. The regularization parameter for R-MDN was set to 0.0001.

## D.4  HAM10000 DATASET

The dataset was first downsampled to $64 \times 64 \times 64$ and then divided into five training stages based on age groups: $< 20$, $[20, 40)$, $[40, 50)$, $[50, 60)$, and $\geq 60$.

- **Stage 1**: 50% of the images came from $< 20$, 30% from $[20, 40)$, 10% from $[40, 50)$, 5% from $[50, 60)$, and 5% from $\geq 60$.
- **Stage 2**: 5% from $< 20$, 50% from $[20, 40)$, 30% from $[40, 50)$, 10% from $[50, 60)$, and 5% from $\geq 60$.
- **Stage 3**: 5% from $< 20$, 5% from $[20, 40)$, 50% from $[40, 50)$, 30% from $[50, 60)$, and 10% from $\geq 60$.
- **Stage 4**: 10% from $< 20$, 5% from $[20, 40)$, 5% from $[40, 50)$, 50% from $[50, 60)$, and 30% from $\geq 60$.
- **Stage 5**: 30% from $< 20$, 10% from $[20, 40)$, 5% from $[40, 50)$, 5% from $[50, 60)$, and 50% from $\geq 60$.

The base model was a ViT with a patch size of 8, 12 hidden layers, 12 heads, a hidden dimension of 384, an MLP dimension of 1536, and the hidden size for the pre-logits layer as 96. Models were trained for 30 epochs with a batch size of 128. Parameters of the R-MDN model were optimized using AdamW (Loshchilov & Hutter, 2017), with a learning rate initialization of 0.0005 that decayed by 0.8 times every 5 epochs. We imposed a weight decay of 0.001. The regularization parameter for R-MDN was set to 0.00001.

# E    ADDITIONAL PLOTS

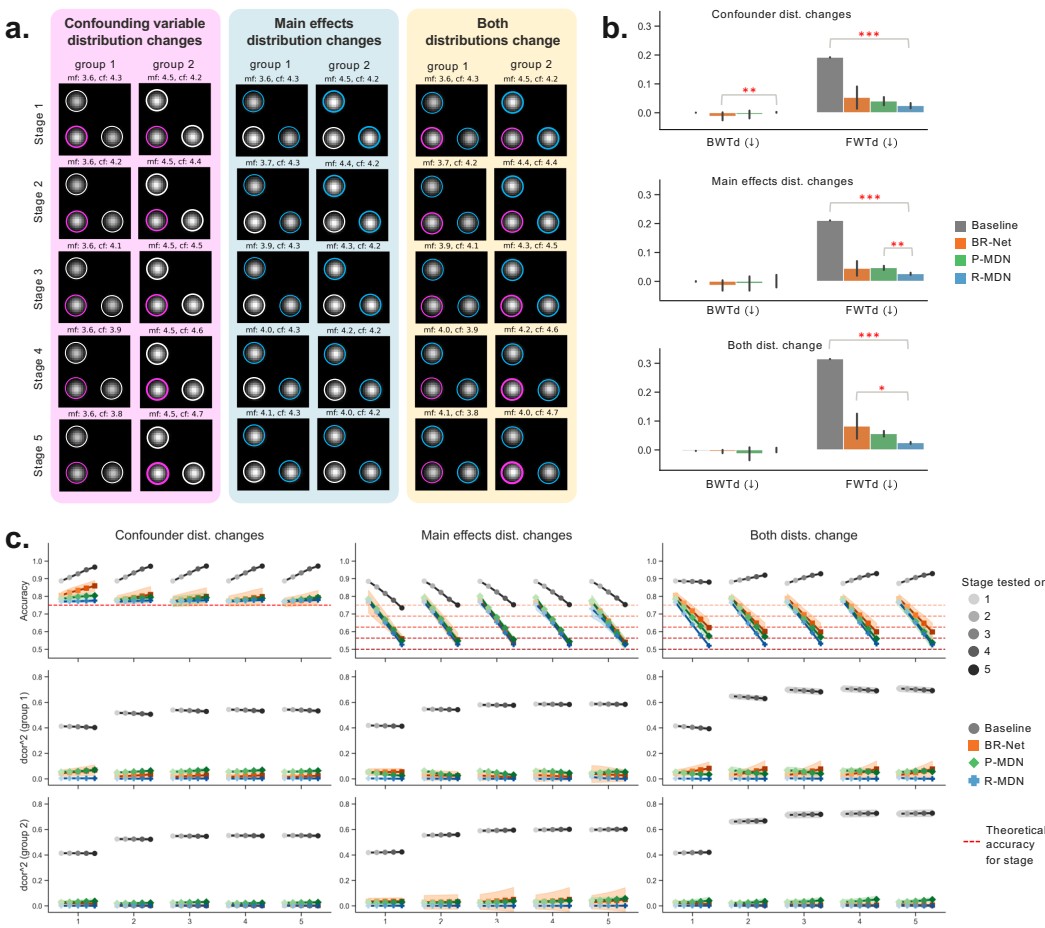

Figure 8: **Synthetic dataset results for continual learning. a.** Samples from the synthetic datasets used for continual learning. We annotate main effects and confounders with boundaries of different widths to visually aid in distinguishing between their magnitudes. **b.** BWTd and FWTd mean and standard deviation for different methods and datasets. The closer the bar to 0, the better the model. A total of 5 runs were performed with different model initialization seeds. A post-hoc Conover's test with Bonferroni adjustment was performed between those groups of methods where a Kruskal-Wallis test showed significant differences ($p < 0.05$). **c.** Accuracy and squared distance correlation for different methods and datasets. For each stage that the model is trained on, it is evaluated against the test sets of all 5 stages (shown through solid curves). Less opaque markers represent earlier stages, while more opaque markers represent later stages being evaluated on. Dotted red lines of various transparency values show the theoretical maximum accuracy that an unbiased model will get for each of the different stages.

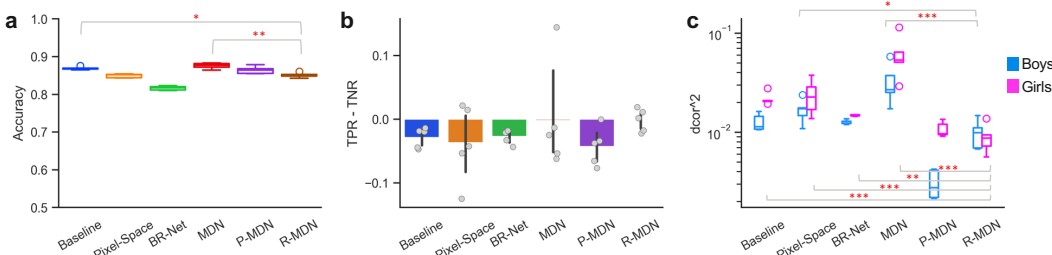

Figure 9: **Visualizing different metrics for the ABCD dataset. a.** Accuracy, **b.** difference between True Positive Rate (TPR) and True Negative Rate (TNR), and **c.** $dcor^2$ between learned features and PDS for boys and girls for different methods. Results shown over 5 folds of 5-fold cross validation, with data split by subject and site ID. Statistically significant differences between R-MDN and other methods are measured first using Kruskal-Wallis and then a post-hoc Conover's test with Bonferroni adjustment.

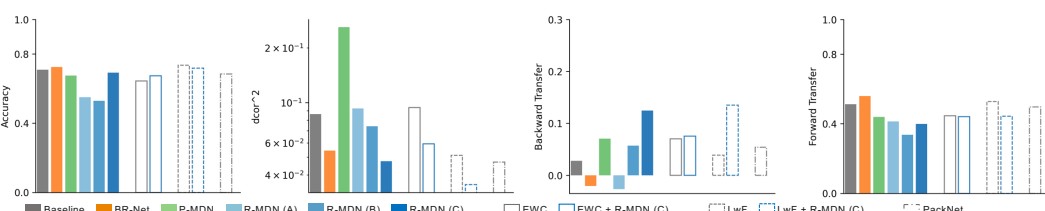

Figure 10: **Visualizing different metrics for HAM10K skin lesion classification.** Accuracy, squared distance correlation, backward transfer, and forward transfer for different methods. Results are shown after training each model on the final training stage.

## F    EFFECT OF REGULARIZATION HYPERPARAMETER

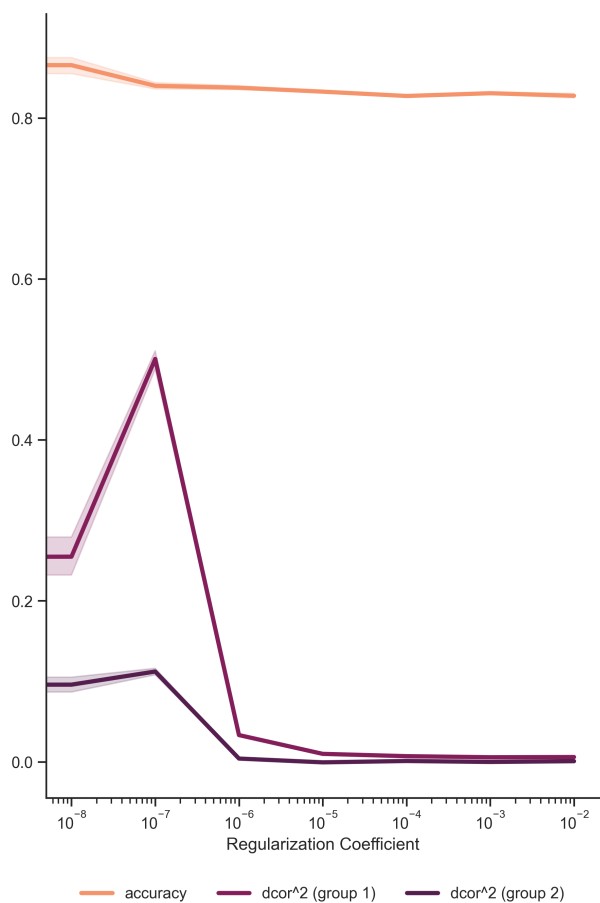

Figure 11: Accuracy and squared distance correlation when the regularization hyperparameter for the R-MDN module is varied. Results are computed for the synthetic dataset described in section 4.1.1, and we show the mean and 95% CI over 100 runs of random model initialization seeds.

In figure 11, we systematically vary the regularization hyperparameter $\lambda$ to assess how sensitive model performance and the ability to learn confounder-independent feature representations are to its value. We observe that model performance remains consistently robust across different values of $\lambda$. However, we find that the capacity to residualize the confounder's effects improves with higher values of $\lambda$, probably due of it stabilizing the residualization process.

# G  R-MDN Module Placement Choice

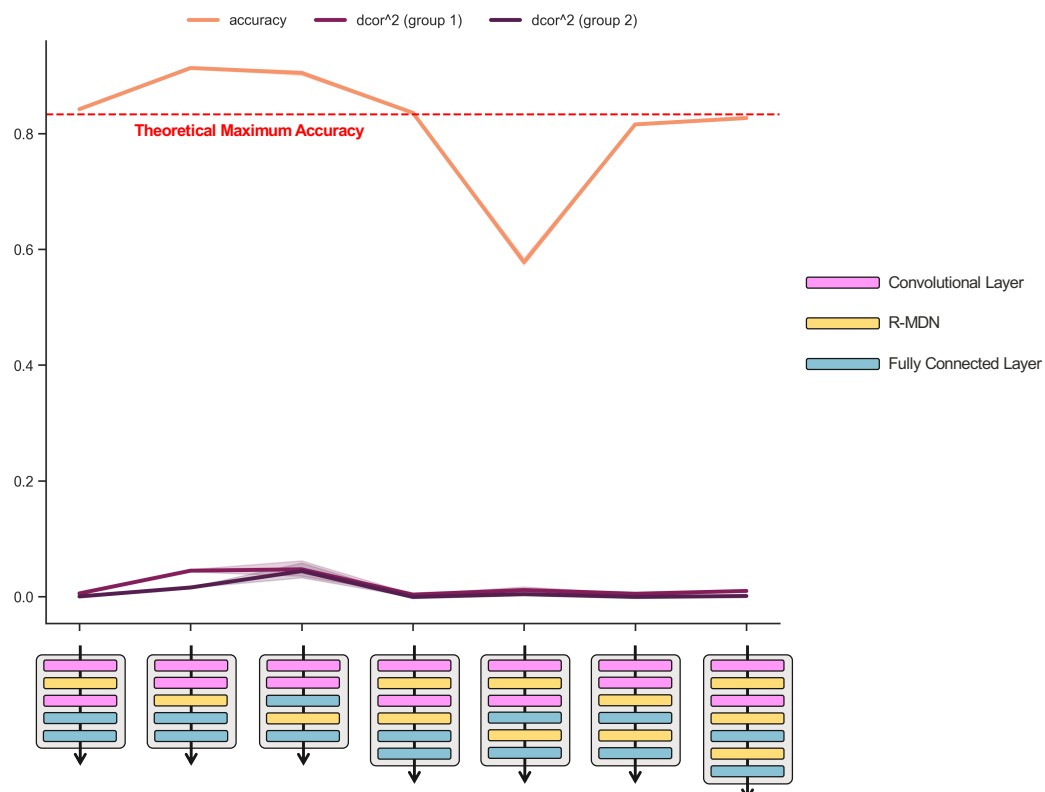

Figure 12: Accuracy and squared distance correlation when the R-MDN module is inserted at various locations in a convolutional neural network. Results are computed for the synthetic dataset described in section 4.1.1, and we show the mean and 95% CI over 100 runs of random model initialization seeds.

In figure G, we vary the placement of the R-MDN layers within a deep convolutional neural network to observe the effects on model performance and correlation of the learned features with the confounder. We find that while model performance seems to be sensitive to the placement, the ability to remove the influence of the confounder from the feature representations is, overall, consistently high. For such an architecture, adding an R-MDN layer after every convolutional layer and the pre-logits layer seems to provide the best trade-off between model performance and residualization (as also observed by Lu et al. (2021)).

# H   GLASS BRAIN VISUALIZATIONS FOR ABCD SEX CLASSIFICATION

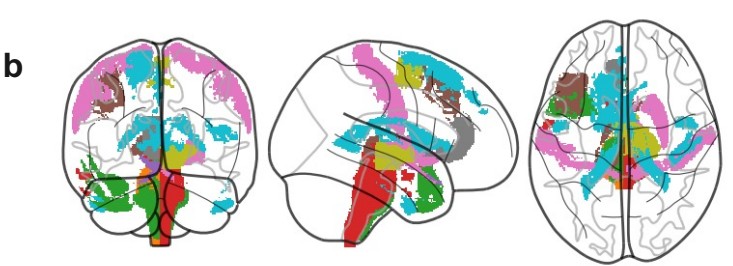

Figure 13: The top 10 regions identified as being relevant for distinguishing sex by **a.** the base model, and **b.** the same model trained with R-MDN.

To identify the top 10 most relevant regions for distinguishing sex (figure H), we first generate 3D saliency maps based on the test set images, highlighting areas in the input image that most activate the model. A threshold of 0.05 is applied to focus on the most salient regions. A 5x5x5 smoothing filter is applied, replacing each voxel's value with the average of its neighboring voxels. These regions are then visualized using the Harvard atlas.

# I  QUANTIFYING SENSITIVITY TO CONFOUNDERS

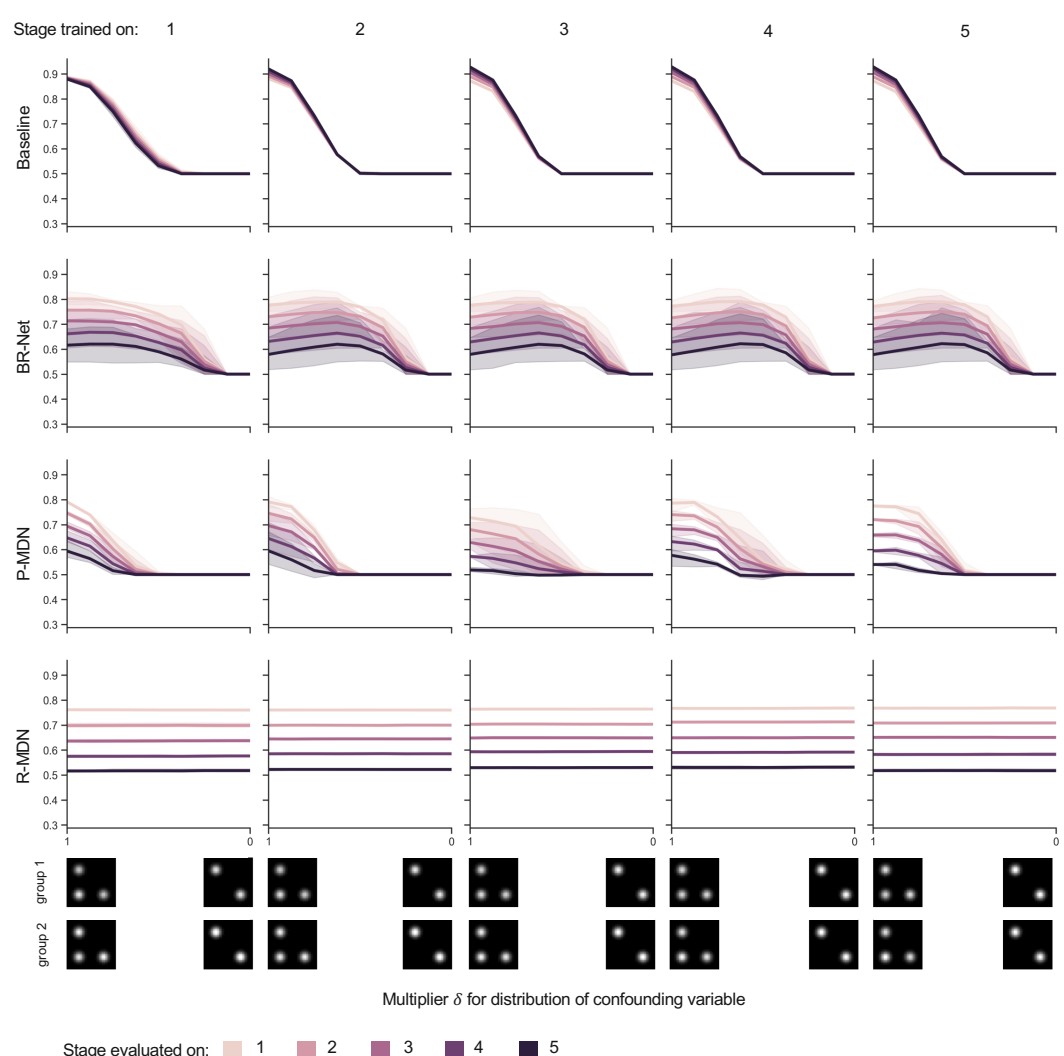

Figure 14: Accuracy that various methods get in a continual learning setting when evaluated on test sets with various distributions of the confounding variable. Each row represents a different method, each column the stage that the model is trained on after which it is evaluated, and each hue of the curve the stage that the model is evaluated on. A $\delta = 0$ implies that that input does not contain a confounder. Results are evaluated on the synthetic dataset from section 4.2.1 where we change the distributions for both the confounding variable and main effets. We show the mean and 95% CI over 3 runs of random model initialization seeds.

In figure 14, we provide additional plots for the experiment visualized in figure 5d. We vary the intensity of the confounder by applying a multiplier $\delta \in [0, 1]$.

## J    EFFECT OF TRAINING PROTOCOL FOR CONTINUAL LEARNING

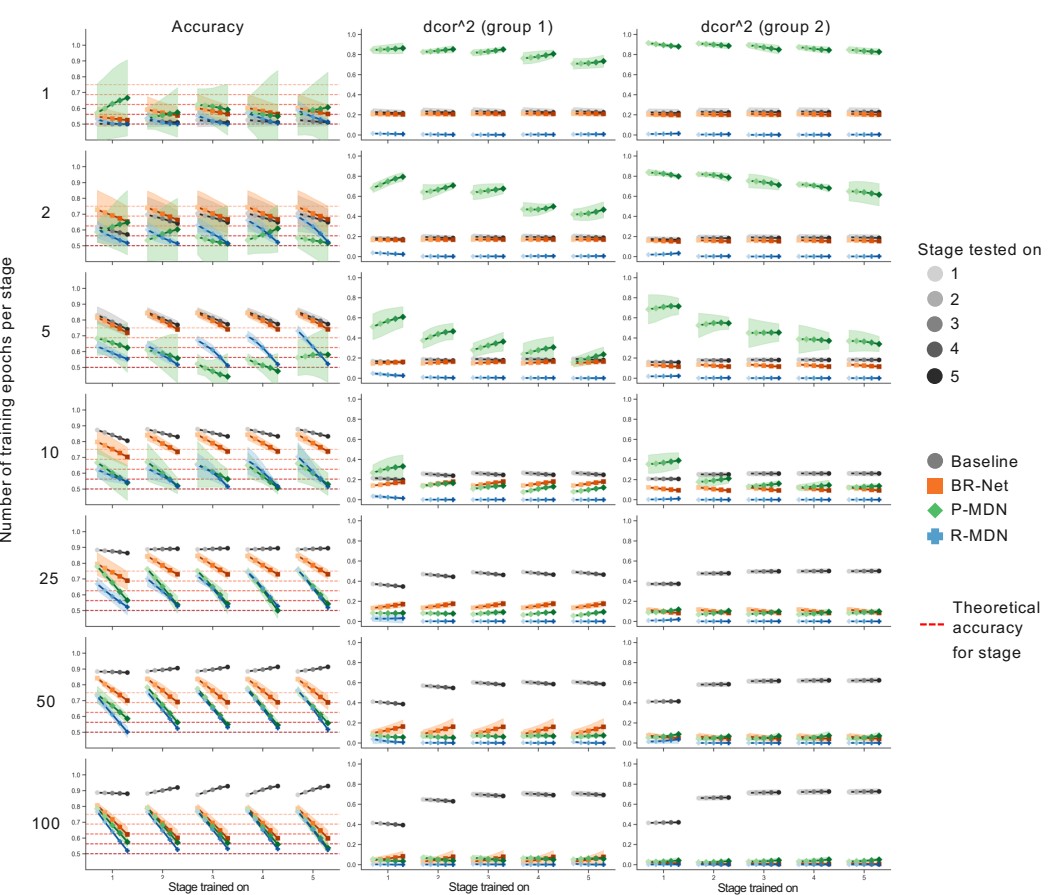

Figure 15: Accuracy and squared distance correlation for different methods and number of training epochs per stage. We used the synthetic dataset where the distributions for both the confounding variable and main effects change. For each stage that the model is trained on, it is evaluated against the test sets of all 5 stages (shown through solid curves). Less opaque markers represent earlier stages, while more opaque markers represent later stages being evaluated on. Dotted red lines show the theoretical maximum accuracy that an unbiased model will get for each of the different stages. Results shown as the mean and 95% CI over 5 runs.

Here, we quantify how task performance and the ability to learn confounder-free feature representations change with different number of training epochs per stage; i.e., with the number of times every example from the training data is presented to the system. We observe that R-MDN is the only method that is able to remove the influence of the confounder from the learned features for smaller number of training epochs. This is perhaps because of R-MDN's fast convergence abilities (Hayes, 1996; Haykin, 2002)—a property that gradient- and adversarial-based methods are not able to demonstrate (figure 15). This is further reinforced by R-MDN having a better forward transfer on future stages of training for both small and large numbers of training epochs (figure 16). Both BR-Net and P-MDN are decent methods for continual learning, but they require the same training examples to be seen multiple times in order to drive high prediction scores and remove confounder influence from learned features.

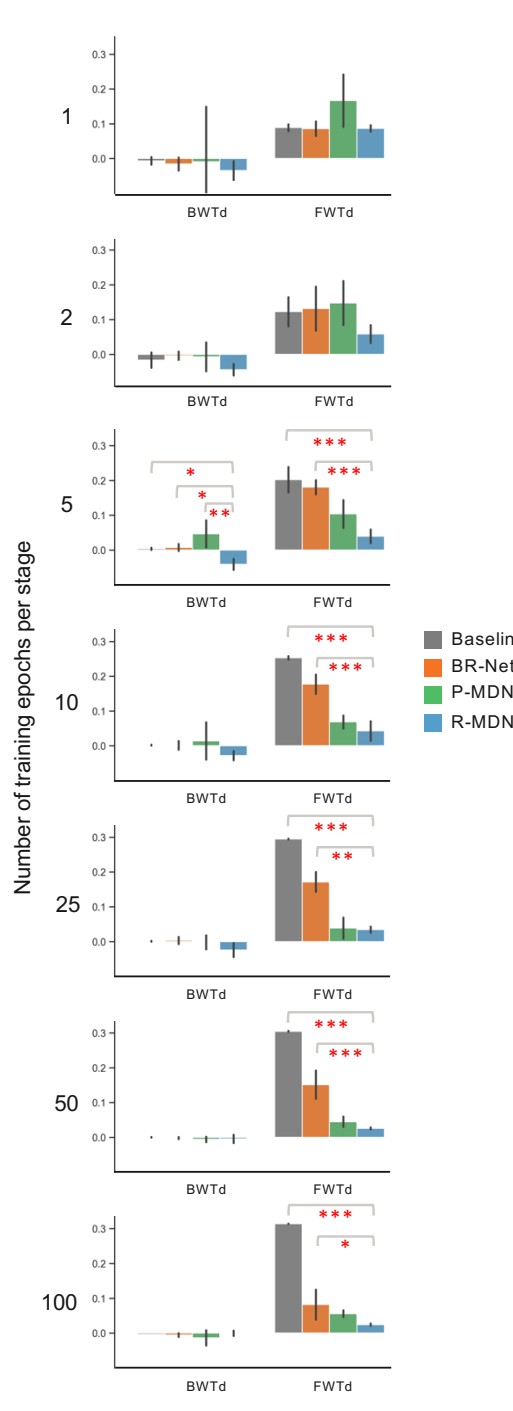

Figure 16: BWT and FWT mean and standard deviation for different methods and number of training epochs per task. We used the synthetic dataset where the distributions of both the confounding variable and main effects change. The closer the bar to 0, the better the model. A total of 5 runs were performed with different model initialization seeds. A post-hoc Conover's test with Bonferroni adjustment was performed between those groups of methods where a Kruskal-Wallis test showed significant differences ($p < 0.05$).

