# OpenReview forum: "Confounder-Free Continual Learning via Recursive Feature Normalization"
_ICLR.cc/2025/Conference — ICLR 2025 Conference Withdrawn Submission_

### Official Review · Reviewer_DDrk · 2024-10-28

**Soundness:** 2
**Presentation:** 4
**Contribution:** 2
**Rating:** 3
**Confidence:** 3

**Summary:**

The paper proposes a new method called Recursive Metadata Normalization (R-MDN) to address the problem of confounding variables in deep neural networks. As confounders influence both the input and the target, this can result in spurious correlations that subsequently distort the true underlying relationships within the data. Furthermore, this can lead to biased and inaccurate predictions, particularly in medical studies where demographic factors like age, sex, and socioeconomic background can affect the outcome. The proposed R-MDN is in the form of a normalization layer that can be inserted at any stage within a NN in order to remove the effects of confounding variables from learned feature representations. It uses the recursive least squares (which has a closed-form solution) to iteratively update its internal parameters based on previously computed values whenever new data is received, allowing the model to adapt dynamically as new data flows in. The novelty of the approach is that R-MDN does not require pre-computation of statistics or batch-level estimates, making it suitable for use with vision transformers and continual learning scenarios. The authors provide a theoretical foundation for their approach, showing how R-MDN can be used to residualize the effects of confounding variables from learned feature representations. They also empirically investigate R-MDN in different experimental setups and NN architectures. In comparison to other state-of-the-art methods, such as MDN and P-MDN, R-MDN shows competitive performance. The paper evaluates R-MDN in longitudinal studies, and continual learning, and shows that R-MDN can help make equitable predictions for population groups, minimizing catastrophic forgetting due to confounders.

**Strengths:**

The paper is very well written and easy to follow. The introduction does a comprehensive analysis of the literature, outlining many previous approaches, highlighting their strengths and weaknesses, as well as motivating the problem at hand. The plots are well formatted (maybe Figure 1 could have a bit larger formula font for readability). Furthermore, the methodology section is clearly written and easy to follow. While going through the formulas I did not find typos. The experimental section is thorough, the experiments are thorough and plots are easily readable.

**Weaknesses:**

I struggle to convince myself about the contribution of the paper and the experimental results. Although the paper is well written and experiments are thorough, when comparing to the original MDN paper, I did not find much novelty in the paper. The proposed usage of working with mini-batches through and using the Sherman-Morrison-Woodbury formula does indeed allow MDN to extend to vision transformers, but I did not find a convincing experiment in the paper that solidifies this claim (also, P-MDN seems to be able to work with vision transformers, although its performance is subpar to that of R-MDN). The theoretical part in Section 3. is clear and easy to follow and does provide a simple idea but I remain unconvinced whether this idea is sufficient for a conference paper.

For example, in the synthetic dataset example, the network which is used is a 2D convolutional neural network, and possibly here is not the right playground where R-MDN would truly shine. From Table 1, we can see that R-MDN has inherited good performance from MDN and that it enjoys favorable performance, however for extremely small batch sizes (2, 16 and 64). The results for batch sizes 256 and 1024 are on parr or worse to MDN. The t-SNE visualisations of learned feature representations do not really tell much - the distribution overlap seems to have just delocalised rather than focused on one region, like in MDN, but that did not really affect my score. For example, does R-MDN perform better or comparatively for batch size 2048 (in the original paper authors use batches 200, 1000 and 2000 so I am curious).

In the ABCD sex classification example, you mention that PDS confounder is significantly larger in girls than in boys but it seems to me that the difference is not significant as the two intervals overlap (please correct me if I am wrong here): 2.175+-0.9 in girls and 1.367+0.6 in boys. The plots in Figure 4 are nice but it is really hard to tell how big is the improvement gain here. Again, the architecture used is a CNN, which I find a bit disappointing considering one of the main motivation for your work is that it can be applied with transformer architectures.

For the continuum of synthetic datasets, although all methods are very good at backward transfer, R-MDN performs better at forward transfer. I find that the plot in 5b (similar to Fig. 4a and 4b) is visually appealing, however it is hard to read how big the improvement is and what the difference is. Why did the authors choose this rather than reporting the results in a Table, akin to Table 1? Figure 5c also very confusingly conveys whether the dist. changes are significant or not, and I am not sure whether it adds much to the paper. I believe here would be beneficial to summarize the benefits of using R-MDN with a transformer architecture to using MDN with non-transformer architecture, and having baselines both for CNNs and vision transformers. The same comment as in previous paragraph goes for the Figure 6. - although the figure is very clean and visually appealing, the accuracy, dcor^2, Backward Transfer and forward transfer are portrayed as a bar chart, and we do not see whether the differences are significant, what is the actual value. It does not really give a convincing argument. Finally, I would say that although the datasets are nicely introduced, the introductions take quite a lot of space and would maybe be more beneficial in the appendix, while the authors could post further experiments, with exact numbers and more comparisons to really highlight whether R-MDN does improve performance and why in the future scientists should resort to using R-MDN rather than P-MDN (which might potentially be hard to tune) or just MDN.

**Questions:**

In line 48 you mention that there is a multitude of situations where some of the previous methods cannot be applied, such as MDN, but you do not mention why is this the case. The question, although it motivates R-MDN well, hangs in the air until the next section. Could you please provide a short explanation of why MDN (and other methods) cannot be applied in this part of the paper?

In line 259, you mention that you apply residualization modules after every convolution and pre-logits layer. Could you briefly explain why did you make this choice? It would be nice to motivate this choice a bit and comment whether other choices might be beneficial for certain applications.

---

> ### Author Response · Authors · 2024-11-19
>
> Thank you for the feedback. Before we respond individually to your concerns, we want to reiterate the contribution of the work. We are not trying to propose a method that beats previous state-of-the-art approaches at visual tasks such as image classification, segmentation, etc. We are also not trying to propose a method for the sake of integration with current state-of-the-art architectures like vision transformers. What we are proposing, however, is a method that performs a visual task (i.e., image classification) while learning fair confounder-free feature representations. In the process, the approach beats previously proposed approaches, can be used with layer-based vision transformers, and describe for the first time applications to continual learning.
>
> ---
>
> > …when comparing to the original MDN paper, I did not find much novelty in the paper.
>
> None of the previous works propose an approach that learns confounder-free representations within the framework of continual learning. This is the biggest novelty.
>
> ---
>
> > a 2D convolutional neural network … is not the right playground where R-MDN would truly shine
>
> We have not designed R-MDN to work with vision transformers. We have designed R-MDN to work within the framework of continual learning with any architecture. Because R-MDN operates on the level of individual examples, it “can be” used in vision transformers, but that is not the focus of this work.
>
> ---
>
> > R-MDN [works] for extremely small batch sizes (2, 16 and 64). The results for batch sizes 256 and 1024 are on parr or worse to MDN.
>
> There are two different things at play here. One, that the size of the dataset is 2048. MDN is *not* better than R-MDN on a batch size of 256 or 1024. It is better than R-MDN on batch sizes that approach the size of the dataset, which is unrealistic for those that contain millions of examples. Second, R-MDN is only worse than MDN by a value of ~0.008  for a batch size of 1024 on the distance correlation metric, which is an insignificant amount.
>
> ---
>
> > The t-SNE visualisations … just delocalised rather than focused on one region
>
> That is the point we are trying to make. What gets delocalized is the distribution of those examples that could belong to either category 1 or 2, but that a biased classifier could still classify into the correct category if they looked at the confounder. R-MDN has this distribution diffused for all batch sizes, meaning that it can never look at confounder information for classification.
>
> ---
>
> > R-MDN with a batch size of 2048
>
> As previously said, a batch size of 2048 here should be analyzed not in its absolute value but that it is the size of the entire training dataset. R-MDN gets a mean accuracy deviation from the theoretical accuracy of 0.173, and an average dcor^2 of 0.008. In contrast, MDN gets a mean accuracy of 0.067 and an average dcor^2 of 0.002.
>
> ---
>
> > PDS difference is not significant
>
> PDS is a significant confounder for the ABCD dataset for sex classification, and the difference for boys and girls is statistically significant, as measured by a two-sample t-test (provided in suppl. C.1). Please also see [1] for a more detailed discussion of the significance of PDS as a confounder in this study.
>
> [1] Ehsan Adeli, Qingyu Zhao, Natalie M Zahr, Aimee Goldstone, Adolf Pfefferbaum, Edith V Sullivan,
> and Kilian M Pohl. Deep learning identifies morphological determinants of sex differences in the
> pre-adolescent brain. NeuroImage, 223:117293, 2020b.
>
> ---
>
> > Some figures are hard to interpret
>
> Thank you for pointing this out. We had originally presented tables for the associated figures in the supplement, but we now moved them into the main text in the updated version of the paper.
>
> ---
>
> > R-MDN with a transformer should be compared against MDN with a non-transformer architecture
>
> While this would be an interesting comparison, like we said earlier, this is not the aim of the work. R-MDN is not designed to simply beat MDN trained using convolutional neural networks. MDN was proposed to work within the framework of static learning, where the model has access to all data at the beginning of training. MDN makes use of this ability, looking at future data to inform current predictions. However, **MDN simply cannot be applied within the framework of continual learning**, due to not having access to future data. Thus, R-MDN naturally beats it through its application. Since our work primarily focuses on continual learning, MDN with any architecture (CNN or ViT) cannot be tested. While the architecture class is secondary to our contribution, we do test R-MDN on both CNNs and ViTs and show that in both a static and continual learning setting, it outperforms prior works valid for the respective settings by learning more fair confounder-free representations.
>
> ---
>
> > Move introduction of the synthetic datasets for continual learning to the supplementary to save space
>
> Thank you, done.

---

> > ### Comment · Reviewer_DDrk · 2024-11-22
> >
> > Thank you for the thorough reply.
> >
> > > We have not designed R-MDN to work with vision transformers. We have designed R-MDN to work within the framework of continual learning with any architecture. Because R-MDN operates on the level of individual examples, it “can be” used in vision transformers, but that is not the focus of this work.
> >
> > Thank you for the clarification, I would then clarify this in the paper as you mention this ability both in the abstract introduction and related work so that was the impression that I got.
> >
> > > That is the point we are trying to make. What gets delocalized is the distribution of those examples that could belong to either category 1 or 2, but that a biased classifier could still classify into the correct category if they looked at the confounder. R-MDN has this distribution diffused for all batch sizes, meaning that it can never look at confounder information for classification.
> >
> > What I meant in my original comment is that t-SNE visualisation is quite unreliable (in my personal opinion) and therefore I was not quite convinced by the experiment. The same goes for the experiments which you have conducted later. I am not an expert in the medical domain so I have therefore reduced my confidence but I still lack the convincing of the novelty of the paper and have therefore decided to keep the score.

---

> > > ### Author Response · Authors · 2024-11-22
> > >
> > > > I would then clarify this in the paper
> > >
> > > Thank you for the comment - we will make this more explicit in the paper.
> > >
> > > ---
> > >
> > > > t-SNE visualisation is quite unreliable (in my personal opinion) and therefore I was not quite convinced by the experiment. The same goes for the experiments which you have conducted later
> > >
> > > We never just rely on t-SNE visualizations or ROI analysis for *any* of our experiments. All experimental results were quantified using accuracy, deviations from theoretical accuracy, distance correlation for measuring the influence of the confounder on the learned features, true and false positive rates, and other metrics. All of these metrics have been used by prior works like MDN, P-MDN, and BR-Net. Qualitative latent feature visualizations are merely auxiliary analyses that become important to try to understand what happens within the system, but they have always been accompanied by quantitative measurements in the paper.
> > >
> > > We then interpret your question as implying that **none of the metrics we have used, or that are used by previously published papers in this space, are reliable or convincing**. We would love to test on metrics in lieu of these to test if a system has learned confounder-free representations if the reviewer knows any and can point us to them. We believe that our quantitative metrics make the experiment convincing, and the qualitative visualizations aim at understanding the system itself better.
> > >
> > > ---
> > >
> > > > I still lack the convincing of the novelty of the paper
> > >
> > > We apologize if we were not very explicit about the novelty of our approach in the paper. We tried to motivate it in the introduction, related works, and discussion sections. We are the **first to propose an algorithm for confounder-free feature representation learning in the domain of continual learning**. If the reviewer believes that this is not novel, we would love to add any citations to previously published work done toward this aim that we might have missed, and the reviewer has in mind.
> > >
> > > And thank you for the feedback on improving our paper!

---

> ### Author Response · Authors · 2024-11-19
>
> > In line 48 you mention that there is a multitude of situations where some of the previous methods cannot be applied, such as MDN, but you do not mention why is this the case. The question, although it motivates R-MDN well, hangs in the air until the next section. Could you please provide a short explanation of why MDN (and other methods) cannot be applied in this part of the paper?
>
> Yes, included in the updated version of the paper. Specifically, none of the prior works have been empirically tested on continual learning data. MDN (1) performs a look-ahead operation to precompute matrix $\Sigma^{-1}$, and (2) uses batch-level estimates in every iteration of training to perform statistical regression. (1) precludes MDN from being applied to continual learning, and (2) precludes it from being used with vision transformers that parallelize operations over individual examples.
>
> ---
>
> > Placement of residualization modules within the architecture
>
> We present the impact of the placement of the modules in section 4.2.2. (for vision transformers) and suppl. G for convolutional neural networks. Deciding where to place the modules is based on empirical performance.

---

### Official Review · Reviewer_FS1M · 2024-11-04

**Soundness:** 2
**Presentation:** 2
**Contribution:** 2
**Rating:** 5
**Confidence:** 2

**Summary:**

The paper presents Recursive Metadata Normalization (R-MDN), a normalization layer designed to reduce the impact of confounders in deep neural networks (DNNs) in both static and continual learning contexts. R-MDN uses the recursive least squares (RLS) algorithm to maintain and update model parameters iteratively. The method demonstrates its effectiveness on both synthetic and real-world datasets.

**Strengths:**

1. The idea of R-MDN is novel and interesting.
2. The idea can be generalized to fields beyond medical context.

**Weaknesses:**

1. This paper spent most of experiment section on the synthetic data part, while experiments on the real data are limited. More experiments and discussions on the real data may make the argument stronger.
2. The experiments on the real world data are not convincing. First, it's a little difficult to read the metric number from the plot --- a table (like table 1) containing accuracy and other related metrics is appreciated. Second, it seems on all the real world experiments, the new method has an accuracy drop. How should we make trade-off here? I think we need more discussions on the benefits this new method could bring, and why the benefits can justify the accuracy drop.
3. What's the computational overhead of the proposed method compared with baseline?

**Questions:**

See weakness section.

---

> ### Author Response · Authors · 2024-11-19
>
> Thank you for your feedback.
>
> ---
>
> > This paper spent most of experiment section on the synthetic data part
>
> In this work, we empirically test our approach in **two different** settings (static and continual learning) and **real medical datasets for both**. With the nature of the work, evaluating the effectiveness of the approach requires ground truth data. With image classification, these are obtained through category labels. However, when dealing with fair confounder-free representations, this becomes non-trivial on real data. We design synthetic simulated environments, carefully accounting for variations in individual variables such as distribution changes in confounders and main effects, so that we can theorize theoretical accuracies (“ground truth” data) to make sense of the results. We take the amount of space we do because we are, to the best of our knowledge, the **first to explore confounder-free representations within continual learning settings**. It thus becomes important to lay the foundation concretely. Even with real medical datasets, we try our best to analyze results by visualizing learned features, ROIs in the brain necessary for sex classification, and comparison with previous biological literature on the topic. We use real medical datasets known to the community, that previous works have built upon. Since this is the first of many more works to come in this space, we hope that future work may be inspired by what we contribute.
>
> ---
>
> > a table (like table 1) containing accuracy and other related metrics is appreciated
>
> Thank you for pointing this out. We had originally presented tables in the supplement, but we moved them into the main text in the updated version of the paper as the reviewer requested.
>
> ---
>
> > more discussions on the benefits this new method could bring
>
> The biggest and most significant benefit of our approach is that it learns confounder-free representations. This becomes important in domains such as medicine, where we would not want to deploy an unfair predictor that is biased toward certain population groups. For example, in section 4.1.2 where we experiment with the ABCD dataset, we show that such a predictors use the cerebellum, which is significantly confounded by PDS, for sex classification, which makes them biased towards girls. R-MDN, on the other hand, does not use information from the cerebellum to make predictions, allowing it to be equitable to populations of either sex.
>
> ---
>
> > and why the benefits can justify the accuracy drop
>
> An accuracy drop does not immediately imply that the model loses its ability to perform task-relevant classification. As demonstrated through experiments on synthetic data, and as a number of previous works such as MDN, P-MDN, BR-Net, etc. show, a biased classifier can inherently obtain a higher classification accuracy by focusing on information it extracts from confounding variables (which it should not). For example, a classifier may estimate an individual has a higher risk for a certain disease based on their sex instead of biomarkers when the biomarkers might be the same for both. The accuracy drop for R-MDN might be due to its ability to not overfit the data and make use of confounder information. It is non-trivial to test if this is exactly the case in real data, which is why synthetic data with ground truth information is important. Additionally, in regards to why this accuracy drop might not be representative of task performance, R-MDN is the only model of those tested that does not show an accuracy drop in the absence of confounders (figure 5 in the updated paper).
>
> ---
>
> > computational overhead of the proposed method compared with baseline
>
> We provide computational and memory complexity in suppl. B of the updated paper.

---

> > ### Comment · Reviewer_FS1M · 2024-11-25
> >
> > Thanks for the reply. I will maintain my score.

---

### Official Review · Reviewer_KFqc · 2024-11-04

**Soundness:** 1
**Presentation:** 3
**Contribution:** 2
**Rating:** 3
**Confidence:** 4

**Summary:**

The work proposes to extend "Metadata Normalization" (Lu et al., 2021) for
continual learning by performing batch-wise updates to the general linear model
solution, specifically using the Sherman-Morrison-Woodbury formula to add newly
observed data to the inverse scatter matrix, where also a regularization term is added.
The static MDN, as well a previous continual version, are compared in two static
experiments. Two more experiments show the performance in a continual setting.

**Strengths:**

- The paper is clearly written.

- The paper provides a broad set of related work.

- Experiments are conducted carefully (with trials and variance in mind).

- The plots are of good quality.

- Various details on the experiments are provided in the appendix.

**Weaknesses:**

- **Missing details in method:** In a continual learning setting, it is true
  that the update of the inverse metadata scatter can be done exactly using the
  Woodbury formula without knowledge of previous data. However, as the $z_i$
  change during training, the $X^T z$ ($Q(N+1)$) in Eq. 1 must be recomputed in
  its entirety. This is in conflict with the continual learning setting, as both
  the required output $z_i$, as well as the metadata $x_i$ of previous data is
  assumed to not be available. Thus, this is not an issue in the input space,
  but only in feature space. I could not find this issue being addressed
  anywhere in the manuscript, but this seems to be the most important part of
  the updated algorithm, as it would otherwise correspond exactly to MDN.

- **Unfocused experiments:** In continual learning, the most interesting
  objective is to quantify the error caused by the constraints of this setting.
  For this work, this means to compare directly with MDN on the full dataset,
  which is done only in Section 4.2 Figure 5b. It would be meaningful to
  directly see the error caused by the iterative updates instead of the
  performance in the static setting. The other experiments in Section 4.2 seem
  to only show this implicitly. The work should focus its empirical analysis
  more on the contribution in continual learning, which I understand as its main
  selling point.


The work proposes an approach to expands MDN to a continual learning setting,
but does so without specifying the important details necessary to do so, but
instead only argues that iterative updates to the metadata scatter inverse using
the Sherman-Morrison-Woodbury formula. The empirical experiments do not show
the error caused by the approximation used in the continual setting.
Without these two main points, I fear that this paper's contribution does not
justify an acceptance.

**Questions:**

- How do you solve the issue with previous data $z_i$ and $x_i$ required to be
  known to compute the residual?

- How large is the error caused by the approximation made by the continual
  setting?

- Is the black bar in Figure 4 plot b the variance? What is the height of each
  block? It would be good to explain this.

- In Figure 4, why does MDN have such a high variance (assuming the bars)? Is this related to the
  small batch size?

---

> ### Author Response · Authors · 2024-11-19
>
> Thank you for the feedback.
>
> ---
>
> > $Q(N+1)$ must be recomputed in its entirety during training
>
> We apologize for not addressing this explicitly in the text. We do not need to recompute $Q$ with every iteration of training, and we present a theoretical derivation of parameter updates in suppl. A of the updated paper. This avoids the reliance on any past (and goes without saying, future) data for making updates to the residual model.
>
> ---
>
> > to quantify the error caused by the constraints of this setting
>
> We have quantified the error due to our approach, and every other prior work that is valid for the setting we work in. We have done so by designing synthetic simulated environments, considering how distributions of confounders, main effects, or both change during continual learning. We theorize theoretical accuracies (“ground truth” data) to compare how R-MDN fares, and this is the required quantification of error imposed by the approach.
>
> ---
>
> > compare directly with MDN on the full dataset
>
> We test every previously proposed algorithm valid for the setting we work in. For static learning, we compare R-MDN against MDN, P-MDN, BR-Net, Pixel-space regularization, etc. For continual learning, however, using MDN does *not* make sense. This is because the very difference between static and continual learning is that you can only train a model on data you have observed so far. This becomes especially important for longitudinal studies, where we might not even have access to the full dataset at the present moment in time. MDN requires a look-ahead operation when it precomputes $\Sigma^{-1}$ with respect to the entire training dataset, which violates the very setting of continual learning. Additionally, even MDN provides sub-par results unless it is trained on close to the entire dataset at once (as mentioned by its authors), i.e., using a batch size that fits the entire dataset. This is why we present results on MDN for static learning but not for continual learning, because MDN does not apply there. We do however compare our approach with P-MDN and BR-Net.
>
> ---
>
> > The work should focus its empirical analysis more on the contribution in continual learning
>
> To the best of our knowledge, we are the first to theorize and empirically demonstrate the effectiveness of a deconfounding approach in continual learning. We design careful datasets, accounting for variations in individual variables, to analyze our approach. We also visualize learned features, regions of interest, deviations from theoretical accuracy, and comparisons with previous works. However, this is not the only setting that R-MDN works in. R-MDN also effectively removes confounder influence in static learning, and that is a domain that the community is also interested in. Providing empirical analysis on static learning helps to compare directly with approaches like MDN that have been adopted before but cannot be applied to continual learning. We believe that our approach being applicable to both domains is an important contribution.
>
> ---
>
> > Figure 4
>
> In the updated paper, we present a table (table 2) quantifying the mean and standard deviation for different approaches to aid interpretability. And yes, the high variance for MDN is related to the batch size used, which is the highest that can be fit in the GPU for training.

---

> > ### Comment · Reviewer_KFqc · 2024-11-25
> >
> > Thank you for your clarifications and updates to the manuscript.
> >
> > ---
> > > $Q(N+1)$ must be recomputed in its entirety during training
> >
> > After reading the new Supplement A, I still do not see this issue addressed.
> > While deriving the updates, the manuscript still seems to simply assume
> > $Q(N+1) = Q(N) + z_{N+1}X_{N+1}$ (Eq. 7), which does not hold as the model
> > changes under training, causing the $z_i, i\in\{1,...,N}$, and therefore also
> > the $Q(N)$ to change. This cannot be computed but only estimated in a continual
> > learning setting, as there is no access to previous data in order to recompute
> > $Q(N)$.
> >
> > Do you simply use the $Q(N)$ without considering its updates?
> >
> > ---
> > > compare directly with MDN on the full dataset
> >
> > I do understand that MDN cannot be used in the continual learning setting on
> > the full dataset, this was not my original concern.
> >
> > However, thinking about it, it is still possible to estimate MDN by simply
> > evaluating on the new data, as your sythetic experiment in Table 1 would
> > suggest, where MDN performs very well with a batch size of 1024. I.e., the
> > statement that MDN requires a batch-size close to the full dataset size does
> > not seem to be supported by your results.
> >
> > Either way, I was mostly interested in the error that is caused by the
> > approximation required the continual setting.
> >
> > ---
> > > Figure 4
> >
> > It should not be an issue to simply accumulate $X^\top X$ as well as $zX$ over
> > multiple batches to get a better estimate. It should not be limited on how much
> > you can fit on your GPU, which gives the wrong impression that MDN
> > underperforms, only due to a technical detail.

---

> > > ### Author Response · Authors · 2024-11-25
> > >
> > > > Do you simply use the Q(N) without considering its updates
> > >
> > > We apologize for misinterpreting your question earlier and thank you for the question. Yes, we do not recompute Q(N) despite the change in learned features of the system. In continual learning, it is not the case that we lose access to previous data; what we do not have is access to future data. A number of different continual learning frameworks, such as replay-based approaches, use prior data to retrain the model and avoid catastrophic forgetting. This, however, adds a computational overhead to the algorithm. When empirically tested, not recomputing Q(N) during the R-MDN update still learns to remove confounder influence from learned features, as the system is run on the same data for multiple epochs. We will include this discussion in the paper.
> > >
> > > ---
> > >
> > > > Error caused by R-MDN with respect to MDN
> > >
> > > Previous literature on the same [1] theorizes that recursive least squares (the linear model constructed by R-MDN) provides an estimate that coincides with the estimate from ordinary least squares (used by MDN) as the amount of data tends to infinity. However, a careful initialization of the inverse covariance matrix (e) can speed up this convergence, as we demonstrate here for a synthetic dataset by providing the $\ell_2$-norm between the estimates of RLS and OLS:
> > >
> > > |  Initialization e | N=1000 | N=2000 | N=3000 | N=4000 | N=5000 | N=6000 | N=7000 | N=8000 | N=9000 | N=10000 | N=11000 | N=12000 | N=13000 | N=14000 | N=15000 | N=16000 | N=17000 | N=18000 | N=19000 | N=20000 |
> > > |--|--|--|--|--|--|--|--|--|--|--|--|--|--|--|--|--|--|--|--|--|
> > > | 0.001 | 2.5979 |2.3925 |2.2206 |2.0697 |1.9413 |1.8309 |1.7292 |1.6380 |1.5569 |1.4820 |1.4172 |1.3511 |1.2963 |1.2453 |1.1985 |1.1571 |1.1159 |1.0778 |1.0416 |1.0096 |
> > > | 0.01 | 2.5958 |1.4086 |0.9737 |0.7424 |0.6020 |0.5095 |0.4388 |0.3837 |0.3417 |0.3062 |0.2795 |0.2544 |0.2358 |0.2194 |0.2045 |0.1934 |0.1819 |0.1719 |0.1626 |0.1558 |
> > > | 0.1 | 2.5805 |0.2953 |0.1656 |0.1165 |0.0884 |0.0736 |0.0614 |0.0506 |0.0431 |0.0362 |0.0322 |0.0282 |0.0261 |0.0241 |0.0215 |0.0210 |0.0193 |0.0181 |0.0168 |0.0170 |
> > > | 0 | 2.5982 |2.5982 |2.5982 |2.5982 |2.5982 |2.5982 |2.5982 |2.5982 |2.5982 |2.5982 |2.5982 |2.5982 |2.5982 |2.5982 |2.5982 |2.5982 |2.5982 |2.5982 |2.5982 |2.5982 |
> > > | 10 | 2.5392 |0.0322 |0.0271 |0.0225 |0.0159 |0.0152 |0.0138 |0.0100 |0.0069 |0.0043 |0.0044 |0.0022 |0.0023 |0.0012 |0.0018 |0.0019 |0.0016 |0.0002 |0.0004 |0.0007 |
> > > | 100 | 2.5380 |0.0280 |0.0251 |0.0211 |0.0147 |0.0143 |0.0133 |0.0096 |0.0066 |0.0043 |0.0044 |0.0023 |0.0023 |0.0013 |0.0021 |0.0021 |0.0018 |0.0004 |0.0007 |0.0005 |
> > > | 1000 | 2.5379 |0.0268 |0.0232 |0.0193 |0.0127 |0.0130 |0.0129 |0.0098 |0.0067 |0.0052 |0.0055 |0.0035 |0.0034 |0.0023 |0.0034 |0.0030 |0.0029 |0.0015 |0.0018 |0.0010 |
> > >
> > > As seen from the table, for a range of hyperparameters such as 10, 100, and 1000, the linear model approximated by R-MDN approaches that of MDN quite quickly, with the difference in estimates being not that significant. Training the model for more samples would have brought the estimates even closer.
> > >
> > > [1] Petre Stoica and Per Åhgren. 2002. Exact initialization of the recursive least-squares algorithm. Int. J. Adapt. Control Signal Process. 16, 3 (April 2002), 219–230. https://doi.org/10.1002/acs.681.
> > >
> > > ---
> > >
> > > > Accumulating statistics across multiple batches for MDN
> > >
> > > Yes, thank you for pointing out. We definitely agree. Like you said, this is a technical detail. R-MDN’s primary motivation is its application to continual learning settings, and we hope that the reviewer would not let this get in the way of the capabilities that R-MDN demonstrates. We can clarify this in the paper as well.

---

> > > > ### Comment · Reviewer_KFqc · 2024-11-26
> > > >
> > > > Thank you for your thoughtful response and the initialization table.
> > > >
> > > > > In continual learning, it is not the case that we lose access to previous data
> > > >
> > > > I respectfully disagree. It is true that some works assume some *replay
> > > > buffer*, which is usually limited in size. However, simply assuming that access
> > > > to the full data remains would mean that there is also no reason to prefer
> > > > R-MDN over MDN, as we can simply recompute $(X^\top X)^{-1}$ and $X^\top z$ over the fully
> > > > accessible dataset until now, without the need to estimate. The updates and batch size
> > > > are simply a technical detail, which can be equally solved through plain MDN
> > > > using OLS.
> > > >
> > > > To continue this thought, we do not even require Sherman-Morrison-Woodbury to
> > > > simply do statistic updates for OLS (by accumulating $X^\top X$ and $X^\top z$), although this is a neat implementation trick.
> > > > This means that rather than introducing a "new" MDN, the focus of the work
> > > > should rather lie more on the statistic learning requirement imposed by the
> > > > continual learning setting for MDN. In the current state, the manuscript is
> > > > formulated as comparing a "new" MDN approach to previous approaches to achieve SOTA
> > > > (which is neither goal-oriented, nor an interesting narration).
> > > > This leaves the continual learning setting somewhat as a benchmark rather than the
> > > > actual main focus of the work.
> > > >
> > > > I find myself in a similar situation as Reviewer DDrk. I still appreciate the
> > > > carefully conducted experiments, but I am afraid the application of the
> > > > Sherman-Morrison-Woodbury to iteratively update the least squares
> > > > solution for MDN alone is rather limited in novelty. I think the work needs some
> > > > serious rewriting to switch its focus, and should be very verbose in what the
> > > > actual limitations are, rather than presenting implementation details as
> > > > short-comings of previous work.
> > > >
> > > > As this requires a considerable effort which is unrealistic to be completed
> > > > during a short rebuttal alone, I will keep my original score.

---

### Official Review · Reviewer_jUcd · 2024-11-05

**Soundness:** 3
**Presentation:** 4
**Contribution:** 2
**Rating:** 5
**Confidence:** 4

**Summary:**

This paper studies an interesting problem of removing the effect of known confounding variables from intermediate feature representations in neural networks. It modifies MDN to deal with sequentially available data by estimating the linear residual model through the recursive least squares method instead of the least squares method used in MDN which needs access to the entire data before the training. Through various experiments, the authors emprically demonstrated that the proposed method R-MDN reduces the confounder effects significantly.

**Strengths:**

The authors resolve the limitations of MDN to deal with sequential data and allow parallel processing of individual input examples. Replacing the least squares method in MDN with the recursive least squares method is a reasonable choice in the sequential nature of data, and it empirically demonstrated the effective removal of confounder effects on the learned features. The authors also visualized the experimental results well.

**Weaknesses:**

Despite the empirical effectiveness, this paper lacks an explanation for the intuition behind the linear residual model and why estimating the coefficient $\beta$ through (recursive) least squares makes sense. Also, RLS provides different solutions from LS even after seeing the same number of samples if we start with $P(0)=\epsilon I$ for $\epsilon\neq0$. Thus, R-MDN and MDN produce different estimations on $\beta$ even for the full-batch data. An explanation of this difference and any (dis)advantages of the solution found by R-MDN compared to that of MDN's solution are missing in the paper.

In the methodology section, some notations are not defined well. ex) $r$ is used for two different residuals. For the second one, $r=z-\tilde{x}\tilde{\beta}_x$ is more natural than $r=\tilde{x}\tilde{\beta}_x-z$ (although the negative sign can be absorbed into the weight parameter of the next layer.) $X_i$ should be explained to be the $i$-th row of $X$.

Currently, the method is written for a single 1-dimensional confounder variable. The same method can be applied to multi-dimensional confounder variables.

**Questions:**

This reviewer wonders about the comparison of computational complexity and memory cost between R-MDN and the baselines, especially MDN and P-MDN.

---

> ### Author Response · Authors · 2024-11-19
>
> Thank you for the feedback.
>
> ---
>
> > this paper lacks an explanation for the intuition behind the linear residual model
>
> In the paper (lines 82-4), we cite prior works that have used statistical regression using linear models to remove the influence of confounders from data. While confounders may interact in a non-linear fashion, in medical settings (as have been considered here and appear frequently in literature), confounders, such as age and sex, almost linearly affect the data. For example, pubertal development scores are higher for girls than boys in early adolescence ages, brain volume is seen to decline with age, etc., which are modeled assuming a linear relationship for the population groups considered within the study. Decisions made from nonlinear models are difficult to interpret, and a sufficiently strong one may decode almost any arbitrary variable from the information present in the features, even if they are not explicitly represented in those features. We would agree that this is a modeling assumption, though adopted extensively in the literature, and the necessity for a nonlinear model such as logistic regression would emerge when the data acted on clearly does not support otherwise. We have updated the paper to present this discussion (lines 84-8).
>
> ---
>
> > comparing estimates from R-MDN and MDN
>
> Prior theoretical work [1] assigns $P(0)$ to $\sum_{i=1}^N (X_i^\top X_i)^{-1}$ to exactly implement the OLS estimate. However, this requires a batch-level update, something we are trying to overcome in this work. The way we initialize $P(0)$ is the most commonly used and the most practical, as it avoids this look-ahead operation. We, as well as all prior works who have used this initialization, accept that the estimate approaches that of the OLS when $N \rightarrow \infty$, which can be mitigated with a careful tuning of $\epsilon$ (a hyperparameter in our work). We have updated the paper with this discussion (lines 201-3).
>
> ---
>
> > Notations
>
> Thank you for pointing them out – we have fixed them in the paper now (lines 178, 188).
>
> ---
>
> > The same method can be applied to multi-dimensional confounder variables.
>
> Medical settings often contain unidimensional confounders, though they can be more than one. Our approach naturally extends to more than one unidimensional confounders by simply column stacking them. Additionally, recursive least squares also supports multi-dimensional confounders as the code we release imposes no assumptions on the dimensionality.
>
> ---
>
> > comparison of computational complexity and memory cost between R-MDN and the baselines
>
> We have updated the paper with this comparison in suppl. B.
>
> ---
>
> [1] Petre Stoica and Per Åhgren. 2002. Exact initialization of the recursive least-squares algorithm. Int. J. Adapt. Control Signal Process. 16, 3 (April 2002), 219–230. https://doi.org/10.1002/acs.681.

---

> > ### Comment · Reviewer_jUcd · 2024-11-26
> >
> > Thank you for the detailed response. I am still not convinced this paper has enough novelty beyond naturally extending MDN by replacing OLS with RLS for the sequential nature of data. Thus, I am keeping my original score. However, I reduced my confidence rate as I am not an expert in the medical AI domain, although I am familiar with the methodology and continual learning.

---

### Author Response · Authors · 2024-11-19
**Revising the paper based on reviewer feedback**

We thank the reviewers for their feedback on our work. We are the first to explore confounder-free representations within continual learning settings, and we are hopeful of the contributions we aim to make in this important space:

- Replacing the least squares method in MDN with the recursive least squares method is a reasonable choice in the sequential nature of data (Reviewer jUcd)

- Experiments are conducted carefully and various details are provided in the appendix (Reviewer KFqc)

- The idea can be generalized to fields beyond medical context (Reviewer FS1M)

- The paper is very well written and the introduction does a comprehensive analysis of the literature (Reviewer DDrk)

---

Based on the feedback we received, we have revised the paper to further improve the way we present our findings. Specifically, we move some of the figures we plot for our experiments into the supplementary and include tables instead for better interpretability of the results. We include a mathematical derivation of the residual model parameters in suppl. A, and the asymptotic computational and memory complexity of R-MDN and other baselines in suppl. B. We also added some minor explanations that the reviewers requested to include in the paper.

---

### Note · Authors · 2024-11-26

**Comment:**

Thank you for your feedback. Unfortunately, the main point of our paper—highlighting how the MDN operation is not applicable to continual learning or even to modern architectures like transformers, as indicated by its original authors and follow-up publications—was missed by the reviewers. We were disheartened to see some reviewers assess our work as if our goal was to improve the SOTA, rather than recognizing that we were introducing a novel concept in continual learning.

During the rebuttal process, we observed a dynamic where reviewers seemed to fall into a negative feedback loop, building on each other’s criticisms while disregarding our responses to those very comments. Some reviewers even shifted their interpretation of the paper’s goal but, instead of adjusting their review scores to reflect this understanding, they lowered their confidence scores! And that happened after the rebuttal.

To avoid further entanglement in this cycle, we have decided to withdraw the paper and focus on refining it for submission to a future venue.

**Withdrawal Confirmation:**

I have read and agree with the venue's withdrawal policy on behalf of myself and my co-authors.